# SONATA: SYNERGISTIC CORESET INFORMED ADAPTIVE TEMPORAL TENSOR FACTORIZATION

**Maolin Wang**[1], **Zhiqi Li**[1,5], **Binhao Wang**[1], **Xuhui Chen**[1], **Tianshuo Wei**[1], **Wanyu Wang**[1],
**Shikai Fang**[2], **Ruocheng Guo**[3], **Zenglin Xu**[4,5,*], **Xiangyu Zhao**[1,*]
[1]City University of Hong Kong, [2]Zhejiang University,
[3]Independent Researcher, [4]Fudan University, [5]Shanghai Academy of AI for Science
[*]Corresponding authors
{morin.wang@,zhiqli6-c@my., xianzhao@}cityu.edu.hk
{xuangufang, rguo.asu, zenglin}@gmail.com

## ABSTRACT

Analyzing dynamic tensor streams is fundamentally challenged by complex, evolving temporal dynamics and the need to identify informative data from high-velocity streams. Existing methods often lack the expressiveness to model multi-scale temporal dependencies, limiting their ability to capture evolving patterns. We propose SONATA, a novel framework that unifies expressive dynamic embedding modeling with adaptive coreset selection. SONATA leverages principled machine learning techniques for efficient evaluation of each observation for uncertainty, novelty, influence, and information gain, and dynamically prioritizes learning from the most valuable data using Bellman-inspired optimization. Entity dynamics are modeled with Linear Dynamical Systems and expressive temporal kernels for fine-grained temporal representation. Experiments on synthetic and real-world datasets show that SONATA consistently outperforms state-of-the-art methods in modeling complex temporal patterns and improving predictive accuracy for dynamic tensor streams. Our code is publicly released at *https://github.com/Applied-Machine-Learning-Lab/ICLR2026_SONATA*.

## 1 INTRODUCTION

Tensors are powerful structures for representing multi-modal data, with applications ranging from recommender systems to neuroscience (Chen et al., 2025; Harshman et al., 1970; Wang et al., 2024b; Sidiropoulos et al., 2017; Wang et al., 2025b;a; 2023a;b;a). In modern scenarios, these tensors often arrive as high-velocity continuous streams (Du et al., 2018; Fang et al., 2021a; 2023). Learning dynamic embeddings from such streams is critical. Yet, two persistent challenges remain: **1) Modeling Expressiveness:** existing approaches often fail to capture the rich and evolving temporal relationships between entities (Zhang et al., 2021; Li et al., 2022; Wang et al., 2022; Chen et al., 2025); and **2) Stream Efficiency:** processing all observations is computationally prohibitive, making it essential to design principled mechanisms for selecting the most informative samples (Wang & Zhe, 2020; Broderick et al., 2013). Addressing both challenges simultaneously is key for advancing streaming tensor learning.

On the modeling side, static methods (Tucker, 1966; Zhe et al., 2016b; Rai et al., 2014) and simple temporal extensions (Xiong et al., 2010; Rogers et al., 2013; Zhe et al., 2016a; Du et al., 2018) suffer from oversimplified temporal representations that cannot capture complex non-stationary dynamics. Even recent dynamic tensor models (Zhang et al., 2021; Fang et al., 2023), though more advanced, still impose restrictive assumptions that limit adaptability to continuously evolving relationships.

On the efficiency side, existing approaches to streaming data typically process everything indiscriminately or adopt heuristic sampling (Broderick et al., 2013; Fang et al., 2021a). This overlooks a central fact: the informational value of streaming observations is highly uneven. Many samples are redundant, while a small fraction is disproportionately important for improving representation quality and predictive accuracy. Without explicitly prioritizing such informative data, models waste computation on low-value observations and risk missing the few points that matter most. This

underscores why principled sample selection is not only desirable but also crucial for accurate and effective streaming tensor analysis.

With these motivations, we introduce **S**ynergistic c**O**reset i**N**formed **A**daptive Temporal **T**ensor f**A**ctorization (**SONATA**), a framework for precise and efficient learning from dynamic tensor streams (Fang et al., 2023; Chen et al., 2025). SONATA is distinguished by two key elements. First, it models fine-grained temporal evolution using Linear Dynamical Systems derived from expressive kernels (e.g., Matérn) (Hartikainen & Särkkä, 2010; Särkkä & Svensson, 2023), enabling the capture of multi-scale dynamics. Second, and most importantly, it introduces a *dynamic coreset strategy* that maintains a compact yet maximally informative subset of the stream. This coreset is updated adaptively by jointly assessing uncertainty, novelty, influence, and information gain, ensuring that the model focuses its updates on the data that matters most.

By aligning expressive modeling with coreset-based efficiency, SONATA provides both the *necessary modeling power* and the *first principled framework to select informative samples in streaming tensor decomposition*. This perspective is at once natural and novel within this field. In contrast, prior work such as (Chhaya et al., 2020) remains confined to symmetric tensor settings, producing static coresets that lack adaptivity and generality, and thus cannot address the challenges of general temporally evolving tensor streams. The main contributions of this work are as follows:

- We propose **SONATA**, which combines dynamic embedding modeling with synergistic coreset construction to handle multi-scale temporal dynamics in tensor streams. This integrated approach improves model expressiveness for accurate temporal pattern modeling.

- We develop a synergistic coreset selection mechanism that evaluates data importance through multiple criteria, i.e., uncertainty, influence, novelty, and information increment, and optimizes coreset composition by principles based on the Bellman equation.

- We develop an efficient coreset-guided streaming Bayesian inference algorithm that leverages the concentrated information for adaptive updates without relying on deep neural networks.

## 2 PROBLEM FORMULATION AND BACKGROUND.

Real-world multiway data can be naturally represented as tensors. Consider a $K$-mode tensor with $d_k$ entities in mode $k$. Each observed entry is indexed by $\boldsymbol{\ell} = (l_1, \ldots, l_K)$, giving dataset $\mathcal{D} = \{(\boldsymbol{\ell}_n, y_n, t_n)\}_{n=1}^N$, where $y_n$ is the value at time $t_n$. The aim is to learn dynamic embeddings $\mathbf{u}_j^{(k)}(t) : \mathbb{R}^+ \to \mathbb{R}^R$ that encode evolving entity properties.

Classical tensor decompositions, such as CP (Harshman et al., 1970) and Tucker (Tucker, 1966), estimate static embeddings. In CP,

$$\mathcal{Y}_{\boldsymbol{\ell}} \approx \sum_{r=1}^R \prod_{k=1}^K u_{l_k,r}^{(k)}, \tag{1}$$

where $u_{l_k,r}^{(k)}$ is the $r$-th entry of the embedding for entity $l_k$ in mode $k$. These approaches ignore temporal evolution. To address nonlinearity, Gaussian processes (GPs) have been used (Xu et al., 2011; Zhe et al., 2015; 2016a), modeling

$$\mathcal{Y}_{\boldsymbol{\ell}} = g(\mathbf{u}_{l_1}^{(1)}, \ldots, \mathbf{u}_{l_K}^{(K)}), \quad g \sim \mathcal{GP}(0, \kappa).$$

Yet inference requires an $N \times N$ kernel matrix, making GPs expensive. With Gaussian noise $\epsilon_n \sim \mathcal{N}(0, \sigma^2)$, the marginal likelihood is $p(\mathcal{Y}) = \mathcal{N}(\mathcal{Y} \mid 0, \mathbf{K} + \sigma^2 \mathbf{I})$.

Temporal information is often handled by adding a time mode or discretizing time (Rogers et al., 2013; Xiong et al., 2010). Dependencies can be modeled conditionally, e.g. $p(\mathbf{u}_{t_{j+1}} | \mathbf{u}_{t_j}) = \mathcal{N}(\mathbf{u}_{t_{j+1}} | \mathbf{u}_{t_j}, \tau^{-1} \mathbf{I})$. Continuous-time variants parameterize CP factors with splines (Zhang et al., 2021), but struggle with irregular sampling or streaming data.

A principled alternative is to model embeddings with Linear Dynamical Systems (LDS). Each entity's latent state $\mathbf{x}_{j,t}^{(k)} \in \mathbb{R}^S$ evolves as

$$\mathbf{x}_{j,t}^{(k)} = \mathbf{F}\mathbf{x}_{j,t-\Delta t}^{(k)} + \mathbf{w}_{j,t}^{(k)}, \quad \mathbf{w}_{j,t}^{(k)} \sim \mathcal{N}(0, \mathbf{Q}), \tag{2}$$

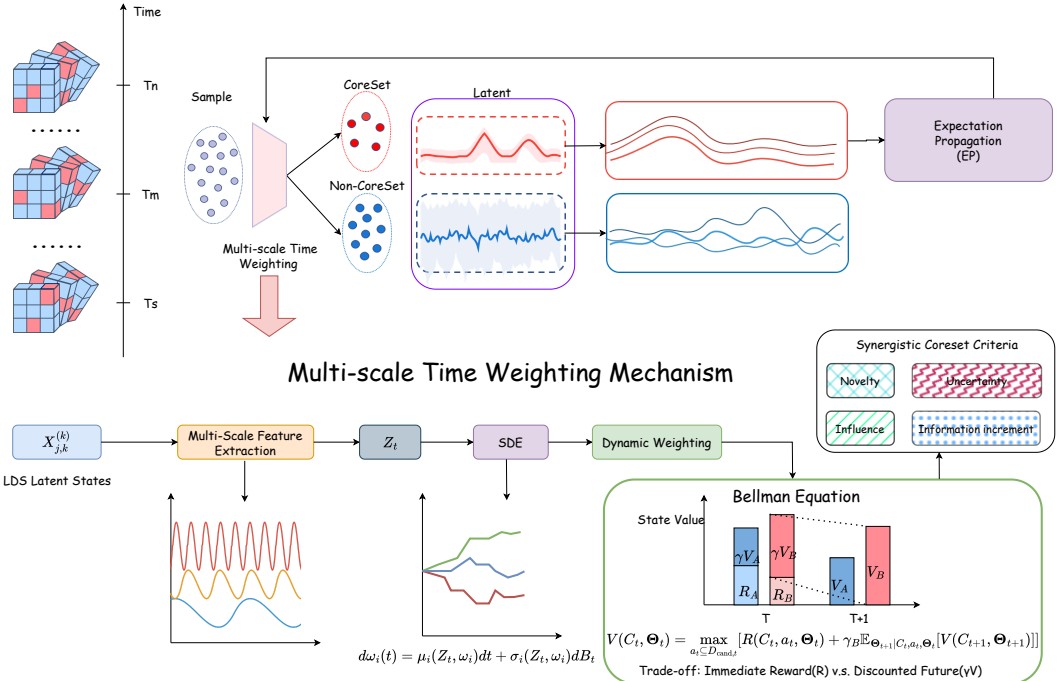

Figure 1: Overview of SONATA framework: Multi-scale feature extraction combined with SDE, synergistic coreset selection criteria (novelty, influence, uncertainty, information increment), coreset evolution via Bellman equation, and optimization via Expectation Propagation.

with observed embedding

$$\mathbf{u}_j^{(k)}(t) = \mathbf{H}\mathbf{x}_{j,t}^{(k)} + \mathbf{v}_{j,t}^{(k)}, \quad \mathbf{v}_{j,t}^{(k)} \sim \mathcal{N}(0, \mathbf{R}_{\text{obs}}). \tag{3}$$

Here, $\mathbf{F}, \mathbf{H}, \mathbf{Q}, \mathbf{R}_{\text{obs}}$ can be linked to continuous-time stochastic differential equations for temporal kernels such as Matérn (Hartikainen & Särkkä, 2010; Fang et al., 2023).

## 3 SONATA MODELS

As shown in Fig. 1, the **S**ynergistic c**O**reset i**N**formed **A**daptive Temporal **T**ensor f**A**ctorization (**SONATA**) is a decomposition model for dynamic tensor streams.

### 3.1 DYNAMIC LATENT FACTOR MODEL WITH TEMPORAL EVOLUTION

At the heart of SONATA lies a model for dynamically evolving latent factors (embeddings) for each entity within the tensor. For a $K$-mode tensor, an entity $j$ in mode $k$ is represented by a time-varying embedding vector $\boldsymbol{u}_j^{(k)}(t) \in \mathbb{R}^R$. The evolution of this embedding is governed by a Linear Dynamical System (LDS), ensuring smooth and continuous trajectories. An underlying latent state $\boldsymbol{x}_j^{(k)}(t) \in \mathbb{R}^S$ (where $S \geq R$) evolves according to the stochastic differential equation (SDE):

$$d\boldsymbol{x}_j^{(k)}(t) = \mathbf{F}\boldsymbol{x}_j^{(k)}(t)dt + \mathbf{L}d\boldsymbol{w}(t), \tag{4}$$

where $\mathbf{F}$ is the dynamics matrix, $\mathbf{L}$ is a noise matrix, and $d\boldsymbol{w}(t)$ is a Wiener process increment with $d\boldsymbol{w}(t) \sim \mathcal{N}(\mathbf{0}, \mathbf{Q}_c dt)$. The observed $R$-dimensional embedding is a linear projection of this state:

$$\boldsymbol{u}_j^{(k)}(t) = \mathbf{H}\boldsymbol{x}_j^{(k)}(t). \tag{5}$$

The parameters $\mathbf{F}$, $\mathbf{H}$, and the steady-state covariance $\mathbf{P}_\infty$ (from which $\mathbf{L}$ and $\mathbf{Q}_c$ can be derived) are determined by a chosen temporal kernel, typically from the Matérn family (Matérn, 1960; Särkkä & Svensson, 2023). For instance, a Matérn-$\nu = 3/2$ kernel implies $S = 2R$, with $\boldsymbol{x}_j^{(k)}(t) =$

$[\boldsymbol{u}_j^{(k)}(t)^\top, \dot{\boldsymbol{u}}_j^{(k)}(t)^\top]^\top$, and specific forms for $\mathbf{F}$ and $\mathbf{H}$. For discrete time steps $\Delta t$, this SDE translates to the discrete-time LDS:

$$\boldsymbol{x}_{j,t}^{(k)} = \mathbf{A}(\Delta t)\boldsymbol{x}_{j,t-\Delta t}^{(k)} + \boldsymbol{w}_{j,t}^{(k)}, \quad \boldsymbol{w}_{j,t}^{(k)} \sim \mathcal{N}(\mathbf{0}, \mathbf{Q}(\Delta t)), \tag{6}$$

$$\boldsymbol{u}_j^{(k)}(t) = \mathbf{H}\boldsymbol{x}_{j,t}^{(k)}, \tag{7}$$

where $\mathbf{A}(\Delta t) = e^{\mathbf{F}\Delta t}$ and $\mathbf{Q}(\Delta t) = \mathbf{P}_\infty - \mathbf{A}(\Delta t)\mathbf{P}_\infty\mathbf{A}(\Delta t)^\top$. An observed tensor entry $y_n$ at time $t_n$ involving entities $\boldsymbol{\ell}_n = (l_{n,1}, \ldots, l_{n,K})$ is modeled, for a CP decomposition, as:

$$y_n = \sum_{r=1}^{R} \prod_{k=1}^{K} u_{l_{n,k},r}^{(k)}(t_n) + \epsilon_n := f(\{\boldsymbol{u}_{l_{n,k}}^{(k)}(t_n)\}_{k=1}^{K}) + \epsilon_n, \tag{8}$$

where $\epsilon_n \sim \mathcal{N}(0, \tau^{-1})$ is observation noise, and $\tau$ is its precision.

## 3.2 SYNERGISTIC CORESET CONSTRUCTION CRITERIA

The computational burden of processing every incoming data point $(\boldsymbol{\ell}_n, y_n, t_n)$ in high-velocity streams necessitates a more efficient approach. SONATA addresses this critical challenge by meticulously maintaining a temporally dynamic coreset $\mathcal{C}_t$. This coreset is not merely a random sample but a compact, dynamically updated subset of all data observed up to time $t$, specifically engineered to be highly informative. The cornerstone of SONATA's coreset strategy is its synergistic selection criteria. Rather than relying on a single heuristic, the inclusion of data points into $\mathcal{C}_t$ is guided by a comprehensive evaluation of their multifaceted potential to refine the model's understanding and enhance its predictive capabilities. This holistic assessment ensures that the coreset captures a rich and diverse representation of the information embedded in the data stream.

This synergy is operationalized through a carefully designed importance score $S_n$ for each candidate data point $n$. This point $n$ is characterized by an observed value $y_n$ at time $t_n$ and involves a set of entities $\boldsymbol{\ell}_n = (\ell_{n,1}, \ldots, \ell_{n,M})$, where $\ell_{n,m}$ is the entity index in mode $m$. The importance score is:

$$S_n = w_u \cdot \mathcal{I}_{\text{unc}}(n) + w_i \cdot \mathcal{I}_{\text{inf}}(n) + w_n \cdot \mathcal{I}_{\text{nov}}(n) + w_m \cdot \mathcal{I}_{\text{mart}}(n). \tag{9}$$

The non-negative weights $w_u, w_i, w_n, w_m$ balance the contributions of the different components.
**Uncertainty.** Here, $\mathcal{I}_{\text{unc}}(n)$ quantifies the model's uncertainty regarding the entities in $\boldsymbol{\ell}_n$ at time $t_n$. Let $\mathbf{V}_{m,\ell_{n,m},t_n}$ be the $R \times R$ predicted covariance matrix of the $R$-dimensional latent embedding $\boldsymbol{u}_{\ell_{n,m}}^{(m)}(t_n)$ for entity $\ell_{n,m}$ in mode $m$ at time $t_n$ (i.e., $\mathbf{V}_{m,\ell_{n,m},t_n} = \text{Cov}(\boldsymbol{u}_{\ell_{n,m}}^{(m)}(t_n)|\mathcal{D}_{t_{n-1}})$). The score is the average of the mean diagonal elements (variances) of these predicted covariance matrices across the $M$ modes:

$$\mathcal{I}_{\text{unc}}(n) = \frac{1}{M} \sum_{m=1}^{M} \left( \frac{1}{R} \sum_{r=1}^{R} [\mathbf{V}_{m,\ell_{n,m},t_n}]_{rr} \right), \tag{10}$$

where $[\mathbf{V}_{m,\ell_{n,m},t_n}]_{rr}$ is the $r$-th diagonal element of the covariance matrix for the embedding of entity $\ell_{n,m}$ in mode $m$. The diagonal elements represent marginal uncertainties of each factor dimension, providing computational efficiency and interpretability while covariance information is implicitly captured through factor interactions in subsequent computations.

**Influence.** $\mathcal{I}_{\text{inf}}(n)$ measures the point's potential influence, typically based on its similarity to members already in the coreset $\mathcal{C}_t$. Let $\boldsymbol{\mu}_{\ell_{n,m},t_n|t_{n-1}}^{(m)}$ be the $R$-dimensional predicted mean embedding of entity $\ell_{n,m}$ in mode $m$ for point $n$. We define an interaction vector for point $n$ as $\mathbf{z}_n = \bigodot_{m=1}^{M} \boldsymbol{\mu}_{\ell_{n,m},t_n|t_{n-1}}^{(m)}$, where $\odot$ denotes the element-wise (Hadamard) product if all embeddings are of the same dimension $R$. This choice mirrors the CP decomposition structure where tensor values are computed as sums of element-wise products of factors, ensuring alignment between influence measurement and the model's prediction mechanism. For a coreset point $c$ (involving entities $\boldsymbol{\ell}_c = (\ell_{c,1}, \ldots, \ell_{c,M})$ at time $t_c$), let $\boldsymbol{\mu}_{\ell_{c,k},t_c|t_c}^{(k)}$ be the posterior mean embedding of entity $\ell_{c,k}$ in mode $k$. Its interaction vector is $\mathbf{z}_c = \bigodot_{k=1}^{M} \boldsymbol{\mu}_{\ell_{c,k},t_c|t_c}^{(k)}$. The similarity $\text{sim}(\mathbf{z}_n, \mathbf{z}_c)$ can be, for example, the cosine similarity: $\text{sim}(\mathbf{z}_n, \mathbf{z}_c) = \frac{\mathbf{z}_n^\top \mathbf{z}_c}{\|\mathbf{z}_n\|\|\mathbf{z}_c\|}$. Then, $\mathcal{I}_{\text{inf}}(n)$ is:

$$\mathcal{I}_{\text{inf}}(n) = \begin{cases} \frac{1}{|\mathcal{C}_t|} \sum_{c\in\mathcal{C}_t} \text{sim}(\mathbf{z}_n, \mathbf{z}_c) & \text{if } \mathcal{C}_t \neq \emptyset \\ 0 & \text{if } \mathcal{C}_t = \emptyset \end{cases}. \tag{11}$$

**Novelty.** $\mathcal{I}_{\text{nov}}(n)$ assesses its novelty compared to existing coreset members $\mathcal{C}_t$. It is a weighted sum:

$$\mathcal{I}_{\text{nov}}(n) = \begin{cases} w_{\text{idx}}\mathcal{I}_{\text{nov,idx}}(n) + w_{\text{time}}\mathcal{I}_{\text{nov,time}}(n) & \text{if } \mathcal{C}_t \neq \emptyset \\ 1 & \text{if } \mathcal{C}_t = \emptyset \end{cases}, \tag{12}$$

where $w_{\text{idx}}$ and $w_{\text{time}}$ are non-negative weights. $\mathcal{I}_{\text{nov,idx}}(n)$ is the proportion of new entity indices in $\boldsymbol{\ell}_n$. Let $E_m(\mathcal{C}_t)$ be the set of unique entity indices from mode $m$ that are present in the coreset $\mathcal{C}_t$.

$$\mathcal{I}_{\text{nov,idx}}(n) = \frac{1}{M} \sum_{m=1}^{M} \mathbb{I}(\ell_{n,m} \notin E_m(\mathcal{C}_t)) \tag{13}$$

where $\mathbb{I}(\cdot)$ is the indicator function (1 if true, 0 if false). $\mathcal{I}_{\text{nov,time}}(n)$ depends on the minimum absolute time difference $\Delta t_{\min}(n) = \min_{c \in \mathcal{C}_t} |t_n - t_c|$ (if $\mathcal{C}_t = \emptyset$, $\Delta t_{\min}(n)$ is treated as $\infty$, making $\mathcal{I}_{\text{nov,time}}(n) = 1$). $\lambda > 0$ is a decay rate hyperparameter.

$$\mathcal{I}_{\text{nov,time}}(n) = 1 - \exp(-\lambda \Delta t_{\min}(n)), \tag{14}$$

**Information increment.** Crucially, $\mathcal{I}_{\text{mart}}(n)$ represents the Martingale-based information increment, estimating the expected reduction in model error (or increase in information) if point $n$ were included. Let $\hat{y}_n$ be the model's prediction for the true value $y_n$, using the predicted mean embeddings $\{\boldsymbol{\mu}_{\ell_{n,m},t_n|t_{n-1}}^{(m)}\}_{m=1}^{M}$. For a CP model of rank $R$, this prediction is $\hat{y}_n = \sum_{r=1}^{R} \prod_{m=1}^{M} [\boldsymbol{\mu}_{\ell_{n,m},t_n|t_{n-1}}^{(m)}]_r$. The term $\Delta E_n$ quantifies the "surprise" or informativeness of point $n$, which can be represented by the squared prediction error:

$$\Delta E_n = (y_n - \hat{y}_n)^2. \tag{15}$$

The Martingale information increment is then:

$$\mathcal{I}_{\text{mart}}(n) = \tanh(\alpha \cdot \max(0, \Delta E_n)), \tag{16}$$

where $\alpha > 0$ is a scaling hyperparameter, and $\tanh(\cdot)$ is the hyperbolic tangent function, which squashes the value, typically into the range $[0, 1)$.

Points with $S_n$ exceeding an adaptive threshold $\theta_t$, potentially combined with an $\epsilon$-greedy exploration strategy, are added to $\mathcal{C}_t$. If $|\mathcal{C}_t|$ exceeds a budget $M_{\max}$, only top$_{M_{\max}}$ will be selected.

## 3.3 TEMPORAL CORESET EVOLUTION VIA BELLMAN EQUATIONS

The decision of which candidate points to include in the coreset at each time step can be framed as a sequential decision-making problem. SONATA employs principles from optimal stopping and dynamic programming, specifically using a Bellman-like equation, to optimize this selection process with respect to long-term model performance.

Let $V(\mathcal{C}_t, \boldsymbol{\Theta}_t)$ be the value function representing the expected future model performance given the current coreset $\mathcal{C}_t$ and model parameters $\boldsymbol{\Theta}_t$. An action $a_t \subseteq \mathcal{D}_{\text{cand},t}$ corresponds to selecting a subset of new candidate points from the candidate set $\mathcal{D}_{\text{cand},t}$ to add to the coreset, resulting in $\mathcal{C}_{t+1} = (\mathcal{C}_t \cup a_t) \setminus \mathcal{P}_t$, where $\mathcal{P}_t$ denotes the set of points pruned to maintain the budget constraint $M_{\max}$. The Bellman equation seeks to maximize the expected cumulative reward:

$$V(\mathcal{C}_t, \boldsymbol{\Theta}_t) = \max_{a_t \subseteq \mathcal{D}_{\text{cand},t}} \left[ \mathcal{R}(\mathcal{C}_t, a_t, \boldsymbol{\Theta}_t) + \gamma_B \mathbb{E}_{\boldsymbol{\Theta}_{t+1}|\mathcal{C}_t, a_t, \boldsymbol{\Theta}_t} [V(\mathcal{C}_{t+1}, \boldsymbol{\Theta}_{t+1})] \right]. \tag{17}$$

The immediate reward $\mathcal{R}(\mathcal{C}_t, a_t, \boldsymbol{\Theta}_t)$ can be defined based on the sum of importance scores $S_n$ of points in $a_t$, or the immediate improvement in model fit or reduction in uncertainty. $\gamma_B \in [0, 1]$ is a discount factor for future rewards. Solving this equation (often approximately, e.g., via lookahead or value function approximation) guides the selection of $a_t$ to maximize long-term utility, rather than just myopic gain. This allows the model to make strategic choices about data retention, potentially prioritizing points that enable better future learning.

### 3.4 BAYESIAN INFERENCE AND ONLINE LEARNING OF SONATA

SONATA employs a streaming Bayesian approach to learn its parameters, primarily the dynamic latent factors (embeddings) $\{\boldsymbol{u}_j^{(k)}(t)\}_{j,k}$ for each entity $j$ in mode $k$ at time $t$, and the observation noise precision $\tau$. The temporal evolution of an embedding $\boldsymbol{u}_j^{(k)}(t) \in \mathbb{R}^R$ is governed by a Linear Dynamical System (LDS) on an underlying latent state $\boldsymbol{x}_j^{(k)}(t) \in \mathbb{R}^S$ (where $S \geq R$), as described by Eq. 6 and Eq. 7. At each timestamp $t_n$, the Kalman filter's prediction step provides a prior distribution $p(\boldsymbol{x}_{j,t_n}^{(k)}|\mathcal{D}_{<t_n})$ for the latent state of entity $j$ involved in the current data, which in turn yields a prior $p(\boldsymbol{u}_j^{(k)}(t_n)|\mathcal{D}_{<t_n})$ for its corresponding embedding.

For an observed tensor entry $(\boldsymbol{\ell}_n, y_n, t_n)$, where $\boldsymbol{\ell}_n = (l_{n,1}, \ldots, l_{n,K})$ are the indices of the involved entities, the observed value $y_n$ is related to their embeddings via a (typically non-linear) function $f(\cdot)$ and Gaussian noise $\epsilon_n \sim \mathcal{N}(0, \tau^{-1})$, such that $y_n = f(\{\boldsymbol{u}_{l_{n,k}}^{(k)}(t_n)\}_{k=1}^K) + \epsilon_n$, as exemplified by the CP decomposition in Eq. 8. Due to the non-linearity of $f(\cdot)$, exact posterior inference is intractable. SONATA thus utilizes Expectation Propagation (EP) to approximate the posterior distributions $p(\{\boldsymbol{u}_{l_{n,k}}^{(k)}(t_n)\}_{k=1}^K, \tau|y_n, \mathcal{D}_{<t_n})$.

Concurrently, the posterior distribution of the noise precision $\tau$ (typically a Gamma distribution with shape $a_\tau$ and rate $b_\tau$) is updated via EP. Its parameters are adjusted based on the expected squared prediction error, $(y_n - \mathbb{E}[f(\{\boldsymbol{u}_{l_{n,k}}^{(k)}(t_n)\}_{k=1}^K)])^2$, and the variance of $f(\cdot)$. The inclusion of a data point in the coreset $\mathcal{C}_{t_n}$ influences its weight in these message updates, with coreset points typically having full weight and non-coreset points potentially having attenuated weights. This mechanism allows SONATA to selectively learn from the most informative data, thereby refining the dynamic embeddings $\boldsymbol{u}_j^{(k)}(t)$ and other model parameters. Overall, by focusing on well-established statistical machine learning techniques rather than computationally expensive deep learning methods, SONATA achieves an effective balance between modeling expressiveness and computational efficiency for streaming tensor factorization. Due to space constraints, the detailed EP update process can be found in **Appendix Sec.** A and our code.

## 4 EXPERIMENTS

In this section, we present experiments for SONATA. Due to space constraints, the implementation details are described in **Appendix Sec.** B.1. Evaluation metrics are presented in **Appendix Sec.** B.2.

### 4.1 SYNTHETIC DATA ANALYSIS

To validate the effectiveness of our method, we begin with a simulation study on synthetic data. A detailed description is provided in Appendix B.3.

We present the estimated factor trajectories from SONATA with a Matérn-3/2 kernel (Fang et al., 2023) and lengthscale 0.3 in Fig. 2. The model successfully recovers the ground-truth trajectories with high accuracy, as shown by the close alignment between estimated and true values. While we acknowledge that tensor decomposition inherently produces non-unique solutions, this non-uniqueness does not diminish the interpretive value of the learned trajectories. Similar to how topic models like LDA provide valuable insights despite non-unique topic assignments, SONATA's trajectories capture meaningful dynamic patterns that serve both interpretive and predictive purposes.

The shaded regions represent the posterior standard deviation, providing quantification of uncertainty for our estimates. Of particular interest is the increased uncertainty at times $t \approx 0.5$, 1, and 1.5–precisely the points where ground-truth trajectories overlap. This demonstrates that SONATA appropriately expresses higher uncertainty when inherent ambiguities exist in the data, providing reliable confidence measures that reflect the true difficulty in distinguishing trajectory values at these time points. This principled uncertainty quantification, enabled by our Bayesian framework, is a key advantage over alternative approaches such as neural networks that may achieve similar predictive performance but lack interpretability.

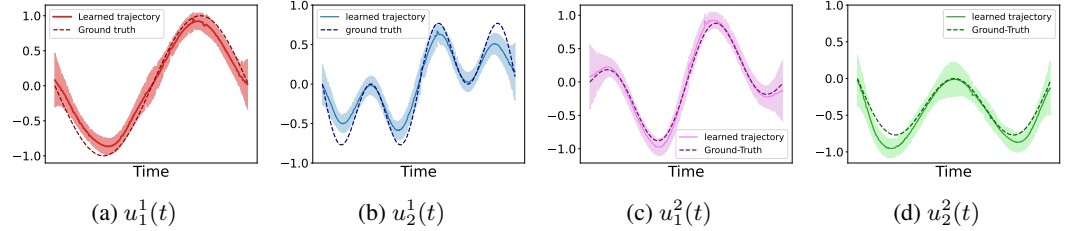

| (a) $u_1^1(t)$ | (b) $u_2^1(t)$ | (c) $u_1^2(t)$ | (d) $u_2^2(t)$ |
|:---:|:---:|:---:|:---:|

Figure 2: The learned factor trajectories from the synthetic data. The shaded region indicates the posterior standard deviation.

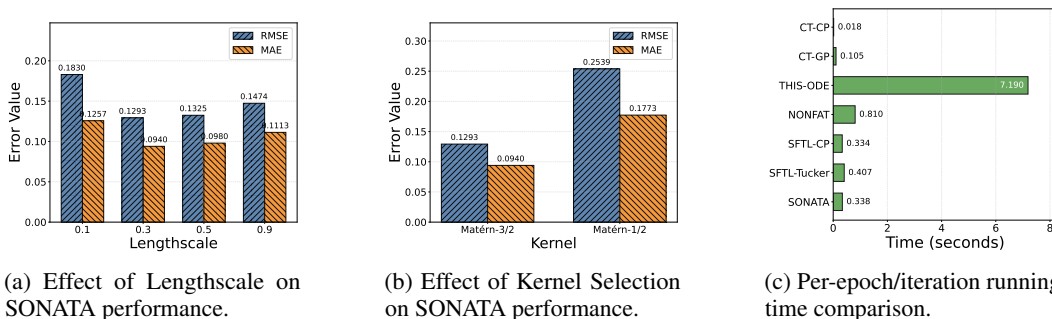

(a) Effect of Lengthscale on SONATA performance.

(b) Effect of Kernel Selection on SONATA performance.

(c) Per-epoch/iteration running time comparison.

Figure 3: Performance analysis of the SONATA model on the Server Dataset.

## 4.2 REAL-WORLD DATA ANALYSIS

**Datasets and Baselines:** Detailed descriptions of the datasets and baseline methods are provided in **Appendix Sec.** B.5 and **Sec.** B.6, respectively.

Our extensive experiments on four real-world datasets demonstrate that SONATA consistently outperforms existing methods. On CA Traffic 30K, SONATA achieved a 61.5% RMSE reduction compared to the second-best method SFTL-CP ($0.231 \rightarrow 0.089$, $p < 0.05$), showing its strength in capturing complex spatiotemporal patterns such as congestion evolution in transportation networks. SONATA also exhibited versatility across diverse domains—environmental monitoring (BeijingAir), infrastructure management (ServerRoom), and user behavior analysis (FitRecord)—validating our combination of Gaussian processes with state-space priors for streaming factor trajectory learning (detailed in Table 1).

Compared with static methods requiring multiple passes and recent continuous-time decompositions, SONATA achieved superior accuracy while processing the data only once (Table 1). This advantage arises from: (1) adaptive updating to evolving patterns versus static assumptions; (2) natural emphasis on recent, more predictive observations; (3) avoidance of overfitting noise common in non-stationary settings; and (4) a coreset mechanism focusing learning on the most informative samples. Against streaming baselines such as POST, ADF-CP, and BASS-Tucker, SONATA maintained substantial advantages throughout, confirming its ability to incrementally build accurate factor trajectories via state-space priors and conditional expectation propagation.

## 4.3 PARAMETER ANALYSIS AND COMPUTATIONAL EFFICIENCY

In our analysis of the lengthscale parameter's effect on model accuracy using the Server dataset (Fig. 3(a)), we found that a lengthscale of 0.3 produces the lowest error rates (RMSE = 0.1293, MAE = 0.0940), indicating this value optimally captures the temporal dynamics in the data. Too small (0.1) or too large (0.9) lengthscales lead to degraded performance due to either overfitting to noise or excessive smoothing of important temporal patterns.

The choice of kernel function significantly impacts model performance. Our comparison (Fig. 3(b)) shows that the Matérn-3/2 kernel substantially outperforms the Matérn-1/2 kernel, reducing RMSE by 49.1% and MAE by 47.0%. This confirms that the Matérn-3/2 kernel, with its moderate smoothness

Table 1: Final prediction error with $R = 5$. The results were averaged from ten runs. **Bold** numbers denote the best performance, underlined numbers represent the second-best results, and $*$ indicates statistical significance at p < 0.05 level using a paired t-test.

| | RMSE | CA Traffic 30K | ServerRoom | BeijingAir | FitRecord |
|---|---|---|---|---|---|
| | PTucker | $0.942 \pm 0.053$ | $0.458 \pm 0.039$ | $0.401 \pm 0.01$ | $0.656 \pm 0.147$ |
| | Tucker-ALS | $1.062 \pm 0.043$ | $0.985 \pm 0.014$ | $0.559 \pm 0.021$ | $0.846 \pm 0.005$ |
| | CP-ALS | $1.093 \pm 0.037$ | $0.994 \pm 0.015$ | $0.801 \pm 0.082$ | $0.882 \pm 0.017$ |
| Static | CT-CP | $0.981 \pm 0.013$ | $0.384 \pm 0.009$ | $0.640 \pm 0.007$ | $0.664 \pm 0.007$ |
| | CT-GP | $0.675 \pm 0.019$ | $0.223 \pm 0.035$ | $0.759 \pm 0.020$ | $0.604 \pm 0.004$ |
| | BCTT | $0.685 \pm 0.024$ | $0.185 \pm 0.013$ | $0.396 \pm 0.022$ | $0.518 \pm 0.007$ |
| | NONFAT | $0.501 \pm 0.002$ | $0.117 \pm 0.006$ | $0.395 \pm 0.007$ | $0.503 \pm 0.002$ |
| | THIS-ODE | $0.632 \pm 0.002$ | $0.132 \pm 0.003$ | $0.540 \pm 0.014$ | $0.526 \pm 0.004$ |
| | POST | $1.004 \pm 0.032$ | $0.641 \pm 0.028$ | $0.516 \pm 0.028$ | $0.696 \pm 0.019$ |
| | ADF-CP | $1.089 \pm 0.041$ | $0.654 \pm 0.008$ | $0.548 \pm 0.015$ | $0.648 \pm 0.008$ |
| Stream | BASS | $1.818 \pm 0.000$ | $1.000 \pm 0.016$ | $1.049 \pm 0.037$ | $0.976 \pm 0.024$ |
| | SFTL-CP | $0.231 \pm 0.015$ | $0.161 \pm 0.014$ | $0.248 \pm 0.012$ | $0.424 \pm 0.014$ |
| | SFTL-Tucker | $0.316 \pm 0.029$ | $0.331 \pm 0.056$ | $0.303 \pm 0.041$ | $0.430 \pm 0.010$ |
| | SONATA (Ours) | $\mathbf{0.089 \pm 0.004}^*$ | $\mathbf{0.115 \pm 0.006}^*$ | $\mathbf{0.237 \pm 0.011}^*$ | $\mathbf{0.414 \pm 0.016}^*$ |
| | MAE | | | | |
| | PTucker | $0.514 \pm 0.006$ | $0.259 \pm 0.008$ | $0.26 \pm 0.006$ | $0.369 \pm 0.009$ |
| | Tucker-ALS | $0.720 \pm 0.006$ | $0.739 \pm 0.008$ | $0.388 \pm 0.008$ | $0.615 \pm 0.006$ |
| | CP-ALS | $0.712 \pm 0.007$ | $0.746 \pm 0.009$ | $0.586 \pm 0.056$ | $0.642 \pm 0.012$ |
| Static | CT-CP | $0.461 \pm 0.003$ | $0.269 \pm 0.003$ | $0.489 \pm 0.006$ | $0.460 \pm 0.004$ |
| | CT-GP | $0.423 \pm 0.001$ | $0.165 \pm 0.034$ | $0.550 \pm 0.012$ | $0.414 \pm 0.001$ |
| | BCTT | $0.452 \pm 0.006$ | $0.141 \pm 0.011$ | $0.254 \pm 0.007$ | $0.355 \pm 0.005$ |
| | NONFAT | $0.391 \pm 0.001$ | $0.091 \pm 0.004$ | $0.256 \pm 0.004$ | $0.341 \pm 0.001$ |
| | THIS-ODE | $0.333 \pm 0.005$ | $0.113 \pm 0.002$ | $0.345 \pm 0.004$ | $0.363 \pm 0.004$ |
| | POST | $0.707 \pm 0.019$ | $0.476 \pm 0.023$ | $0.352 \pm 0.022$ | $0.478 \pm 0.014$ |
| | ADF-CP | $0.904 \pm 0.007$ | $0.496 \pm 0.007$ | $0.385 \pm 0.012$ | $0.449 \pm 0.006$ |
| Stream | BASS | $1.601 \pm 0.041$ | $0.749 \pm 0.01$ | $0.934 \pm 0.037$ | $0.772 \pm 0.031$ |
| | SFTL-CP | $0.026 \pm 0.001$ | $0.108 \pm 0.008$ | $\mathbf{0.150 \pm 0.003}$ | $0.242 \pm 0.006$ |
| | SFTL-Tucker | $0.177 \pm 0.005$ | $0.216 \pm 0.034$ | $0.185 \pm 0.029$ | $0.246 \pm 0.001$ |
| | SONATA (Ours) | $\mathbf{0.015 \pm 0.001}^*$ | $\mathbf{0.083 \pm 0.004}^*$ | $0.156 \pm 0.011$ | $\mathbf{0.240 \pm 0.012}^*$ |

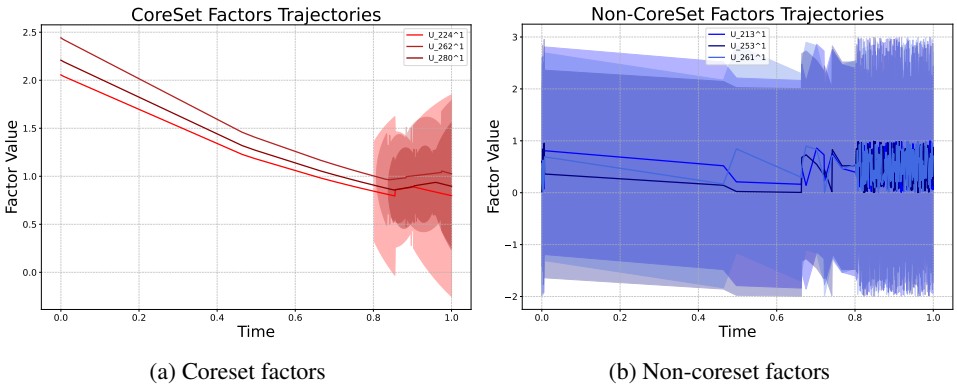

(a) Coreset factors          (b) Non-coreset factors

Figure 4: Comparison of temporal patterns between coreset and non-coreset factors. Coreset factors in (a) exhibit more structured and consistent behavior with clearer patterns, while non-coreset factors in (b) display more irregular and noisy trajectories.

properties, better captures the underlying patterns in spatiotemporal tensor data. Computational efficiency is crucial for practical applications alongside accuracy.

Our runtime comparison of different methods (Fig. 3(c)) demonstrates that SONATA delivers superior performance with reasonable computational cost. While simpler methods like CT-CP execute faster (0.018s per iteration), they deliver significantly lower accuracy as evidenced in Table 1. In contrast, THIS-ODE achieves reasonable accuracy but demands substantially more computation time (7.190s per iteration) due to its deep neural architectures. SONATA, with a computation time of 0.338s per iteration, achieves the highest accuracy across all datasets, demonstrating excellent effectiveness without deep neural networks or excessive computational burden for streaming tensor tasks.

## 4.4 COMPARISON OF CORESET AND NON-CORESET FACTOR TRAJECTORIES

As shown in Fig. 4, the temporal patterns of selected coreset and non-coreset factors demonstrate distinct characteristics. The coreset factors exhibit more structured and consistent behavior with clearer patterns, while the non-coreset factors display more irregular and noisy trajectories. It is important to note that these trajectories represent one interpretation of the system dynamics rather than absolute truths–neural network models could provide entirely different yet valid interpretations. However, SONATA's interpretation offers crucial advantages: the clear behavioral patterns in coreset factors (e.g., periodicity suggesting regular events like daily backups) versus the high uncertainty in non-coreset factors provide valuable signal-noise distinction for domain experts. This contrast highlights SONATA's ability to effectively identify and select the most informative and representative factors from the dataset. The temporal priors imposed by our LDS and Matérn kernel further constrain the solution space toward smooth, temporally continuous trajectories, yielding stable and meaningful patterns that directly contribute to our superior predictive performance shown in Table 1. Discussions about the trajectories of all coresets and servers can be found in **Appendix Sec.** C.1.

## 4.5 CORESET BUDGET AND SIZE ANALYSIS

A key parameter in our proposed procedure is the maximum coreset size, $M_{\max}$, which serves as a budget constraint. A legitimate question then seems to be: How can this budget be defined and to what extent does it impact the performance? In order to alleviate this, we conducted a systematic sensitivity analysis on the CA Traffic dataset. The findings, found in Table 2, highlight the core insight of the adaptive approach of our algorithm.

Table 2: Sensitivity analysis of the coreset budget ($M_{\max}$) on the CA Traffic dataset. The results demonstrate that the algorithm automatically converges to an effective coreset size without necessarily utilizing the full budget.

| $M_{\max}$ | Final RMSE | Peak Memory (MB) | Avg Update Time (ms) | Final Coreset Size | Coreset Usage (%) |
|---|---|---|---|---|---|
| 1000 | 0.1808 | 7.84 | 8897.64 | 800 (2.67%) | 80.0% |
| 2000 | 0.0938 | 8.12 | 3553.56 | 1597 (5.32%) | 79.9% |
| 3000 | 0.0891 | 8.18 | 3536.70 | 1654 (5.51%) | 55.1% |

Experimental results show that our algorithm does not blindly fill the budget but rather autonomously converges to an optimal coreset size. For instance, if we increase $M_{\max}$ from 2000 to 3000, we can see the saturation phenomenon. Though the budget grew by $50\%$, the final coreset size increased only narrowly from 1597 to 1654, and the utilization rate declined from 79.9% to 55.1%. The improvement in RMSE was just $5\%$. This means those 1600 high-quality samples are adequate to obtain the necessary underlying data stream information. Over the same threshold, the marginal efficiency of adding new ones decreases considerably. This saturation behavior is initiated by the four collaborative selection criteria interacting. The novelty decreases, uncertainty is reduced, and errors converge as the coreset grows, since it is now a representative set of entities and timestamps. This results in an automatic increase in the implicit inclusion threshold; that is, only observations with significant informational value are included in the coreset.

## 4.6 EFFECT OF DISCOUNT FACTOR

The discount factor $\gamma$ in the Bellman equation balances immediate vs. future rewards in coreset selection. Table 3 shows the RMSE of SONATA on the Server and CA Traffic datasets. For Server, $\gamma = 0.5$ yields the lowest RMSE (0.1156), indicating immediate rewards dominate. For CA Traffic, $\gamma = 0.9$ performs best (0.0893), meaning long-term coreset utility is more useful. This highlights the data-dependent nature of $\gamma$. Due to space limitations, additional analysis about hyperparameter and coreset can be found in **Appendix Sec.** C.

Table 3: RMSE performance with different discount factors.

| Discount Factor | Server | CA Traffic |
|---|---|---|
| 0.9 | 0.1293 | 0.0893 |
| 0.7 | 0.1181 | 0.1072 |
| 0.5 | 0.1156 | 0.1107 |
| 0 | 0.1409 | 0.1716 |

## 5 RELATED WORKS

**Temporal Tensor Decomposition and Streaming Methods.** Traditional tensor decomposition methods Battaglino et al. (2018); Wang et al. (2019; 2020); Bader & Kolda (2008) handle static data but lack temporal dynamics and require multiple passes. Early works treated time as an additional mode Rogers et al. (2013), while recent methods like CT-CP Zhang et al. (2021), CT-GP Chen et al. (2024), BCTT Fang et al. (2022), and trajectory-based models (e.g., THIS-ODE Li et al. (2022), NONFAT Wang et al. (2022)) capture continuous evolution. LDS also models temporal relations Zhen et al. (2023), but these methods require full datasets and multi-epoch training, making them unsuitable for high-velocity streams. Streaming methods such as POST Du et al. (2018), ADF-CP Wang & Zhe (2020), and BASS-Tucker Fang et al. (2021a) update CP/Tucker factors incrementally. OnlineGCP Phipps et al. (2023) extends CP to exponential family distributions but does not model dynamics over continuous time. SOFIA Lee & Shin (2021) provides seasonal modeling but requires a pre-set seasonal cycle, while our Matérn kernel learns multi-scale temporal patterns. OR-MSTC Najafi et al. (2019) handles streaming tensors focusing on spatial growth, whereas SONATA captures temporal evolution. SBDT Fang et al. (2021b) uses deep neural networks, but their black-box nature makes temporal patterns harder to interpret compared to SONATA's factor trajectories, which provide intuitive insights.

**Coreset Strategies for Tensor Learning.** General coreset theory Langberg & Schulman (2010) has inspired tensor-specific sampling, including LineFilter and KernelFilter for streaming contractions Chhaya et al. (2020), Bayesian regression coresets Huggins et al. (2016), Lewis weights Cohen & Peng (2015), randomized/decomposition Battaglino et al. (2018), tensor sketching Song et al. (2016); Wang et al. (2015), and RandNLA for matricized tensors Song et al. (2019). However, such methods rely on local criteria, overlooking evolving dynamics and long-term utility. Streaming tensor approaches with GP/LDS or ODEs remain computationally heavy, as state complexity grows with data. SONATA advances this by jointly measuring uncertainty, influence, novelty, and information gain, while optimizing long-term benefit via Bellman principles—making it, to our knowledge, the first coreset-based streaming tensor decomposition that fully integrates temporal considerations.

## 6 CONCLUSION

We presented SONATA, a unified framework for streaming tensor factorization that integrates expressive continuous-time modeling with a synergistic coreset selection strategy. By leveraging linear dynamical systems derived from temporal kernels, SONATA captures complex, multi-scale temporal dynamics of entities. Its coreset mechanism dynamically selects informative data points based on uncertainty, influence, novelty, and information gain, and optimizes long-term utility via Bellman-inspired principles. Our online Bayesian inference algorithm further ensures efficient and adaptive updates. While SONATA demonstrates strong empirical performance, several limitations remain. First, the current implementation assumes Gaussian observation noise and linear dynamical systems derived from Matérn kernels, which may restrict modeling flexibility in certain non-Gaussian or highly nonlinear domains. Second, our method assumes streaming data arrives at consistent temporal intervals; performance under bursty or irregular stream patterns remains to be fully explored. Due to space constraints, LLM usage details are provided in **Appendix Sec** D.

## ACKNOWLEDGMENTS

This research was partially supported by National Natural Science Foundation of China (No.62502404), Hong Kong Research Grants Council (Research Impact Fund No.R1015-23, Collaborative Research Fund No.C1043-24GF, General Research Fund No.11218325), Institute of Digital Medicine of City University of Hong Kong (No.9229503), Huawei (Huawei Innovation Research Program), Tencent (Tencent Rhino-Bird Focused Research Program, Tencent University Cooperation Project), Alibaba (CCF-Alimama Tech Kangaroo Fund No. 2024002), Didi (CCF-Didi Gaia Scholars Research Fund), Kuaishou (CCF-Kuaishou Large Model Explorer Fund, Kuaishou University Cooperation Project), and Bytedance.

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

# A    DETAILS OF EXPECTATION PROPAGATION ALGORITHM IN SONATA

The core challenge in SONATA's learning process is to infer the posterior distribution of the dynamic latent embeddings $\{\boldsymbol{u}_{l_{n,k}}^{(k)}(t_n)\}_{k=1}^K$ given an observation $y_n$ at time $t_n$ and all previous data $\mathcal{D}_{<t_n}$. Due to the non-linear relationship $y_n = f(\{\boldsymbol{u}_{l_{n,k}}^{(k)}(t_n)\}_{k=1}^K) + \epsilon_n$ (as in Eq. 8), this posterior is intractable. EP addresses this by iteratively refining an approximation to the true posterior, typically within the exponential family (e.g., Gaussian).

## A.1    EP UPDATE FOR DYNAMIC LATENT EMBEDDINGS

The EP algorithm approximates the true likelihood term $p(y_n|\{\boldsymbol{u}_{l_{n,k}}^{(k)}(t_n)\}_{k=1}^K, \tau)$ with simpler, tractable site approximations (often called approximate factor messages). When updating the parameters for a specific embedding $\boldsymbol{u}_{l_{n,k'}}^{(k')}(t_n)$ of entity $l_{n,k'}$ in mode $k'$, we aim to compute the parameters of this site approximation, which we will refer to as the "message" from the likelihood factor concerning $y_n$ to the variable $\boldsymbol{u}_{l_{n,k'}}^{(k')}(t_n)$. This message encapsulates the information that the observation $y_n$ provides about $\boldsymbol{u}_{l_{n,k'}}^{(k')}(t_n)$, effectively marginalizing out the other embeddings $\{\boldsymbol{u}_{l_{n,k}}^{(k)}(t_n)\}_{k\neq k'}$ and the noise precision $\tau$ using their current estimates (i.e., their current posterior predictive distributions).

This message is chosen to be a Gaussian distribution. For the Canonical Polyadic (CP) decomposition, the observation model is $y_n \approx \langle \boldsymbol{u}_{l_{n,k'}}^{(k')}(t_n), \boldsymbol{w}_{\backslash k',n}\rangle + \epsilon_n$, where $\boldsymbol{w}_{\backslash k',n} = \bigodot_{j\neq k'} \boldsymbol{u}_{l_{n,j}}^{(j)}(t_n)$ is the element-wise product of embeddings from modes other than $k'$ for observation $n$. The likelihood factor is $p(y_n|\boldsymbol{u}_{l_{n,k'}}^{(k')}(t_n), \{\boldsymbol{u}_{l_{n,j}}^{(j)}(t_n)\}_{j\neq k'}, \tau) = \mathcal{N}(y_n|\boldsymbol{u}_{l_{n,k'}}^{(k')}(t_n)^\top \boldsymbol{w}_{\backslash k',n}, \tau^{-1})$. The Gaussian message approximating this factor with respect to $\boldsymbol{u}_{l_{n,k'}}^{(k')}(t_n)$ has natural parameters: a precision matrix $\boldsymbol{\Lambda}_{\mathrm{msg},k'}$ and a mean-precision product (also called information vector) $\boldsymbol{\eta}_{\mathrm{msg},k'}$. These are derived as:

$$\boldsymbol{\Lambda}_{\mathrm{msg},k'} = \mathbb{E}[\tau] \cdot \mathbb{E}_{\{\boldsymbol{u}_j\}_{j\neq k'}}\left[\boldsymbol{w}_{\backslash k',n}\boldsymbol{w}_{\backslash k',n}^\top\right], \tag{18}$$

$$\boldsymbol{\eta}_{\mathrm{msg},k'} = \mathbb{E}[\tau] \cdot y_n \cdot \mathbb{E}_{\{\boldsymbol{u}_j\}_{j\neq k'}}\left[\boldsymbol{w}_{\backslash k',n}\right]. \tag{19}$$

The expectations $\mathbb{E}_{\{\boldsymbol{u}_j\}_{j\neq k'}}$ are taken with respect to the current posterior distributions of the embeddings $\{\boldsymbol{u}_{l_{n,j}}^{(j)}(t_n)\}_{j\neq k'}$ (obtained from their respective Kalman filters at time $t_n$ prior to this update iteration), and $\mathbb{E}[\tau]$ is the current expectation of the noise precision (from its Gamma posterior). Damping is often applied when updating these message parameters from one EP iteration to the next to improve convergence stability. If an entity $l_{n,k'}$ participates in multiple observations at the current time $t_n$, the natural parameters ($\boldsymbol{\Lambda}_{\mathrm{msg},k'}$ and $\boldsymbol{\eta}_{\mathrm{msg},k'}$) from each such observation are summed to form an aggregated message for that entity.

This aggregated Gaussian message, now characterized by $\boldsymbol{\Lambda}_{\mathrm{agg},k'}$ and $\boldsymbol{\eta}_{\mathrm{agg},k'}$, is then converted to moment parameters: mean $\boldsymbol{\mu}_{\mathrm{pseudo},k'}$ and covariance $\boldsymbol{V}_{\mathrm{pseudo},k'}$, to serve as a pseudo-observation for the Kalman filter:

$$\boldsymbol{V}_{\mathrm{pseudo},k'} = (\boldsymbol{\Lambda}_{\mathrm{agg},k'})^{-1}, \tag{20}$$

$$\boldsymbol{\mu}_{\mathrm{pseudo},k'} = \boldsymbol{V}_{\mathrm{pseudo},k'}\boldsymbol{\eta}_{\mathrm{agg},k'}. \tag{21}$$

The Kalman filter tracks the latent state $\boldsymbol{x}_{l_{n,k'}}^{(k')}(t_n)$, from which the embedding is derived via $\boldsymbol{u}_{l_{n,k'}}^{(k')}(t_n) = \mathbf{H}\boldsymbol{x}_{l_{n,k'}}^{(k')}(t_n)$. The filter's prediction step provides the prior distribution for the state at $t_n$ based on data up to $t_{n-1}$ (or the last time this entity was updated, $t_{prev}$): $p(\boldsymbol{x}_{l_{n,k'}}^{(k')}(t_n)|\mathcal{D}_{<t_n}) = \mathcal{N}(\boldsymbol{x}_{l_{n,k'}}^{(k')}(t_n)|\boldsymbol{m}_{\mathrm{x,prior}}, \boldsymbol{P}_{\mathrm{x,prior}})$. Specifically, $\boldsymbol{m}_{\mathrm{x,prior}} = \mathbf{A}\boldsymbol{m}_{\mathrm{x,post}}(t_{prev})$ and $\boldsymbol{P}_{\mathrm{x,prior}} = \mathbf{A}\boldsymbol{P}_{\mathrm{x,post}}(t_{prev})\mathbf{A}^\top + \mathbf{Q}$, where $\mathbf{A}$ is the state transition matrix, $\mathbf{Q}$ is the process noise covariance, and $\boldsymbol{m}_{\mathrm{x,post}}(t_{prev}), \boldsymbol{P}_{\mathrm{x,post}}(t_{prev})$ are the posterior mean and covariance from the previous update of this entity. The Kalman filter incorporates the pseudo-observation ($\boldsymbol{\mu}_{\mathrm{pseudo},k'}, \boldsymbol{V}_{\mathrm{pseudo},k'}$)

using its standard update equations:

$$\text{Innovation: } \boldsymbol{\nu}_n = \boldsymbol{\mu}_{\text{pseudo},k'} - \mathbf{H}\boldsymbol{m}_{\text{x,prior}}, \tag{22}$$

$$\text{Innovation Covariance: } \mathbf{S}_{\text{KF},n} = \mathbf{H}\boldsymbol{P}_{\text{x,prior}}\mathbf{H}^\top + \boldsymbol{V}_{\text{pseudo},k'}, \tag{23}$$

$$\text{Kalman Gain: } \mathbf{K}_{\text{KF},n} = \boldsymbol{P}_{\text{x,prior}}\mathbf{H}^\top \mathbf{S}_{\text{KF},n}^{-1}, \tag{24}$$

$$\text{Updated State Mean: } \boldsymbol{m}_{\text{x,post}} = \boldsymbol{m}_{\text{x,prior}} + \mathbf{K}_{\text{KF},n}\boldsymbol{\nu}_n, \tag{25}$$

$$\text{Updated State Covariance: } \boldsymbol{P}_{\text{x,post}} = \big(\mathbf{I} - \mathbf{K}_{\text{KF},n}\mathbf{H}\big)\boldsymbol{P}_{\text{x,prior}}. \tag{26}$$

This yields the updated posterior for the latent state, $p(\boldsymbol{x}_{l_{n,k'}}^{(k')}(t_n)|\mathcal{D}_{\leq t_n}) = \mathcal{N}(\boldsymbol{x}_{l_{n,k'}}^{(k')}(t_n)|\boldsymbol{m}_{\text{x,post}}, \boldsymbol{P}_{\text{x,post}})$. Consequently, the updated posterior for the embedding is $p(\boldsymbol{u}_{l_{n,k'}}^{(k')}(t_n)|\mathcal{D}_{\leq t_n}) = \mathcal{N}(\boldsymbol{u}_{l_{n,k'}}^{(k')}(t_n)|\mathbf{H}\boldsymbol{m}_{\text{x,post}}, \mathbf{H}\boldsymbol{P}_{\text{x,post}}\mathbf{H}^\top)$. This iterative EP process (cycling through factors and variables) refines the estimates of all involved embeddings for the current timestamp $t_n$.

## A.2 EP UPDATE FOR NOISE PRECISION $\tau$

The observation noise precision $\tau$ is also learned via EP. SONATA typically assumes a Gamma prior for $\tau$, $p(\tau) = \text{Gamma}(\tau|a_0, b_0)$, where $a_0$ is the shape and $b_0$ is the rate parameter. The likelihood term $p(y_n|\{\boldsymbol{u}_{l_{n,k}}^{(k)}(t_n)\}_{k=1}^K, \tau) = \mathcal{N}(y_n|f(\cdot), \tau^{-1})$ also depends on $\tau$.

To update the posterior $p(\tau|\mathcal{D}_{\leq t_n})$, which remains a Gamma distribution $\text{Gamma}(\tau|a_N, b_N)$, EP considers the contribution of each observation $y_n$. The message from the likelihood factor $p(y_n|\cdot, \tau)$ to $\tau$ effectively updates the parameters of the Gamma posterior. The shape parameter $a_N$ is typically updated by adding $1/2$ for each observation processed. The rate parameter $b_N$ is updated by adding $\frac{1}{2}\mathbb{E}_{\{\boldsymbol{u}\}}[(y_n - f(\{\boldsymbol{u}_{l_{n,k}}^{(k)}(t_n)\}_{k=1}^K))^2]$. This expectation is taken with respect to the current posteriors of the embeddings. It can be approximated as $\frac{1}{2}\big((y_n - \mathbb{E}[f(\cdot)])^2 + \text{Var}[f(\cdot)]\big)$, where $\mathbb{E}[f(\cdot)]$ is the expected prediction and $\text{Var}[f(\cdot)]$ is its variance, both computed using the current embedding posteriors. This process accumulates evidence about the noise level from each data point.

## A.3 INFLUENCE OF THE CORESET

It is important to note that the coreset mechanism $\mathcal{C}_{t_n}$ influences these EP updates. Data points selected into the coreset typically contribute with full weight to the message calculations and subsequent posterior updates. Conversely, data points not in the coreset might have their influence attenuated (e.g., their messages are down-weighted). This strategy allows SONATA to focus its learning capacity on the most informative observations, thereby efficiently refining the dynamic embeddings $\boldsymbol{u}_j^{(k)}(t)$ and other model parameters like $\tau$ in a streaming fashion.

## A.4 CORESET SELECTION PROCESS

Not all high-scoring points in a batch are automatically selected for the coreset. Each point is evaluated independently against the selection criteria. When multiple points exceed the threshold, they compete for the limited coreset budget ($M_{\text{max}}$). Only the top-scoring points up to the budget limit are retained, ensuring computational efficiency while capturing the most informative samples.

## A.5 MULTI-SCALE FEATURE EXTRACTION

Multi-scale feature extraction is fundamental to SONATA and can be realized using both the Matérn kernel and LDS. We build on the Matérn-3/2 kernel, which in our application operates in a state space of dimension $2R$. This kernel is used in conjunction with the embeddings $(u(k)_j(t))$ and their derivatives $(\dot{u}(k)_j(t))$. This augmented state representation is useful for encoding instantaneous variants as either the derivatives to the embeddings or long-term trends, i.e. embeddings themselves as embeddings. With both the embedding values and derivatives together, SONATA chooses between fast and slow overflowing oscillation based on fast versus slow, and in this way, it can compute a multi-scale temporal dynamics. Moreover, the time novelty nature of SONATA has an exponential

time-decay function. Also the data decay with $\lambda$ parameter has direct effects on data temporal quality. Older data points lose their impact and become less important, allowing the model to focus on more recent data, thereby maintaining the temporal aspect, but allows for a more overarching structure like a timeline.

## B   MORE EXPERIMENT SETTINGS

### B.1   IMPLEMENTATION DETAILS

The SONATA model was implemented with PyTorch (Paszke et al., 2019), TensorLy (Kossaifi et al., 2019), and TedNet (Pan et al., 2022), and run on an Intel Core Ultra 7 155H CPU. For all real-world datasets, we used CP decomposition with embedding dimension $R = 5$. Dataset-specific configurations were as follows: for the traffic dataset, we employed a Matérn-1/2 kernel with lengthscale 0.9, discount factor 0.9, evaluation interval 10, and coreset maximum size of 3000; for the Beijing dataset, a Matérn-3/2 kernel with lengthscale 0.3, discount factor 0.1, evaluation interval 20, and coreset size of 100; for the Server dataset, a Matérn-3/2 kernel with lengthscale 0.3, discount factor 0.5, evaluation interval 60, and coreset size of 400; and for the fitRecord dataset, a Matérn-1/2 kernel with lengthscale 0.1, discount factor 0.5, evaluation interval 6, and coreset size of 2000. It is worth noting that the configurations detailed above were selected to achieve the optimal performance reported in the main results (Table 1).

For the ablation studies and parameter sensitivity analyses, we adopted a fixed baseline configuration to strictly control variables and isolate the impact of specific factors. In particular, for the Server dataset, these analytical experiments were consistently conducted using a discount factor of 0.9 to observe relative trends under a standardized setting, unless otherwise specified.

For the synthetic data experiments, we adopted a Matérn-3/2 kernel with lengthscale 0.3 and embedding dimension $R = 2$. The model was trained for 100 epochs with dataset-specific evaluation intervals, utilizing martingale-based dynamic coreset selection with importance weights $[0.3, 0.2, 0.2, 0.3]$.

For runtime comparisons, it is important to note the fundamental differences between static and streaming methods. Static methods like CP-ALS require multiple passes through the entire dataset and, following their original papers and standard practice, we set fixed iteration counts (e.g., 100 iterations). Their total processing time far exceeds SONATA, as they were not designed for streaming scenarios. In contrast, streaming methods process data once in a single pass, similar to SONATA.

Fig. 3c presents per-iteration/epoch runtime comparisons for representative methods. While simpler streaming approaches like CT-CP demonstrate faster per-iteration times, their predictive accuracy is substantially lower than SONATA's (as shown in Table 1). Methods pursuing comparable high accuracy levels, such as THIS-ODE, incur much higher computational costs (7.190s per iteration) compared to SONATA (0.338s per iteration) while still achieving lower predictive accuracy. This demonstrates that SONATA successfully balances efficiency with superior performance.

### B.2   EVALUATION METRICS

To comprehensively evaluate the performance of SONATA and baseline methods on dynamic tensor streams, we adopt the following widely used metrics.

**Root Mean Square Error (RMSE).** RMSE measures the square root of the average squared differences between the predicted and true tensor entry values. It is defined as

$$\text{RMSE} = \sqrt{\frac{1}{N}\sum_{n=1}^{N}(\hat{y}_n - y_n)^2}, \tag{27}$$

where $N$ is the number of evaluated entries, $y_n$ is the ground-truth value, and $\hat{y}_n$ is the predicted value. Lower RMSE indicates higher predictive accuracy.

**Mean Absolute Error (MAE).** MAE calculates the average absolute difference between predicted and true values:

$$\text{MAE} = \frac{1}{N}\sum_{n=1}^{N}|\hat{y}_n - y_n|, \tag{28}$$

MAE is more robust to outliers compared to RMSE and reflects the typical prediction deviation.

These metrics jointly quantify prediction accuracy, robustness, and efficiency, providing a comprehensive basis for evaluating the effectiveness of SONATA in streaming tensor factorization tasks.

### B.3 GENERATED TRAJECTORIES

We generated a two-mode tensor with two nodes per mode, where each node is represented by a time-varying factor trajectory. The factor trajectories for the first mode were defined as $u_1^1(t) = \sin(2\pi t)$ and $u_2^1(t) = \cos(2\pi t)\sin(4\pi t)$, while for the second mode, they were $u_1^2(t) = \sin(3\pi t)\cos(\pi t)$ and $u_2^2(t) = \sin(2\pi t)\sin(\pi t)$. Given these factors, tensor entry values at time $t$ were generated via $y_{(i,j)}(t) \sim \mathcal{N}(u_i^1(t)^T u_j^2(t), 0.01)$. We randomly sampled 500 timestamps from the interval $[0.5, 1.5]$ and, for each timestamp, selected two tensor entries with values sampled according to the model above, resulting in 1,000 observed values in total. The near-zero uncertainty before $t = 0.8$ occurs because the trajectories are well-separated in that region and we have dense observations providing strong evidence. The smooth kernel prior is well-suited to these underlying trigonometric functions, leading to high confidence. Our experiments focus on the prediction of future observations based on streaming history, demonstrating SONATA's ability to track and predict complex temporal patterns in tensor data.

### B.4 HYPERPARAMETER SELECTION GUIDELINES

There are some key hyperparameters that impact performance in SONATA and the way they are chosen can affect much as an effect on outcome. For **coreset budget (Mmax)** we recommend that this be 5%-10% of expected data stream size as upper bound. This makes the adaptive coreset automatically converge toward the optimal size, making this a budget guideline rather than a hard limit. But as the coreset size falls outside of this range, performance starts to reduce and computational overhead gradually increases even further for high-throughput coresets.

For the **lengthscale** of the temporal kernel, our experiments show us that a range between 0.3 and 0.5 leads to stable performance, with RMSE variations staying below 3%. The decision to use values outside of this range can negatively influence model performance either due to overfitting noise given a small lengthscale, or excessively smoothing important temporal features given a large lengthscale.

The **kernel selection** is also important, where the Matérn-3/2 kernel is the appropriate chosen kernel. This shows that it performs relatively better than the Matérn-1/2 kernel in capturing the multiscale temporal dynamics, therefore making the model much more accurate.

When it comes to the $\lambda_{diversity}$ parameter, we prefer around 0.1 to 0.3 for most datasets. This bounds a trade-off with a diversity and richness in the coreset: for example, allowing the model select appropriate samples from an extensive corpus of data without introducing excess data points that will overwhelm the model with irrelevant samples. Similarly, the $\tau_{novelty}$ parameter should be adjusted based on the sparsity of data; we would recommend 0.5 for sparse data and 0.9 for dense data, where higher values would favor novel and important data points.

The **discount factor ($\gamma$)** of the Bellman equation that considers early or later rewards in coreset selection runs best in a range of 0.5 to 0.9 based on data. For immediate rewards-only datasets $\gamma = 0.5$ is most robust, and for long-term utility-oriented sets $\gamma = 0.9$ is more preferable. For example, when it comes to the CA Traffic dataset $\gamma = 0.9$ is the right choice, but Server data outperforms with $\gamma = 0.5$.

### B.5 DATASETS

We evaluated SONATA on four real-world temporal tensor datasets. 1) **CA Traffic 30K** (Moosavi et al., 2019) contains lane-blocked records in California from January 2018 to December 2020, extracted as a three-mode temporal tensor between 5 severity levels, 20 latitudes, and 16 longitudes. Different from many existing papers, we adopted a more complex setup with 30K entry values and their timestamps. [1]. 2) **FitRecord Dataset** is a collection of outdoor exercise health logs from EndoMondo users' health status, structured as a three-mode tensor encompassing 500 users, 20

---
[1]https://smoosavi.org/datasets/lstw

sports types, and 50 altitude levels. The tensor entries represent heart rates, with 50,000 timestamped observations recorded. [2]. 3) **ServerRoom Dataset** contains temperature logs from the Poznan Supercomputing and Networking Center (Niwiński et al., 2003), organized as a three-mode tensor consisting of 3 air conditioning modes, 3 power usage levels (50%, 75%, 100%), and 34 locations. The dataset contains 10,000 timestamped temperature readings. [3]. 4) **BeijingAir Dataset** includes air pollution measurements in Beijing from 2014 to 2017 (Song et al., 2017), structured as a two-mode tensor between monitoring sites and pollutants ($12 \times 6$ dimensions). The dataset includes 20,000 timestamped concentration measurements. [4].

### B.6 BASELINES

For evaluation, we compared SONATA against a set of tensor baselines. The static methods require multiple data passes and include: PTucker (Oh et al., 2018), a parallel Tucker decomposition using row-wise updates; Tucker-ALS (Bader & Kolda, 2008) and CP-ALS (Battaglino et al., 2018), utilizing alternating least squares for Tucker and CANDECOMP/PARAFAC decomposition respectively; CT-CP (Zhang et al., 2021), a continuous-time CP decomposition with polynomial splines; CT-GP (Chen et al., 2024), employing Gaussian processes to model tensor entries as functions of latent factors and time; BCTT (Fang et al., 2022), a Bayesian continuous-time Tucker decomposition that models the tensor-core as a time-varying function; NONFAT (Wang et al., 2022), which employs nonparametric factor trajectory learning in the frequency domain; and THIS-ODE (Li et al., 2022), utilizing neural ODEs to model entry values. We also evaluated against streaming methods that process data in a single pass: POST (Du et al., 2018), a probabilistic streaming CP decomposition using mean-field variational Bayes; ADF-CP (Wang & Zhe, 2020), combining assumed density filtering with conditional moment matching; BASS-Tucker (Fang et al., 2021a), which employs online sparse tensor-core estimation via spike-and-slab priors; and SFTL-CP/Tucker (Fang et al., 2023), representing streaming factor trajectory learning with CP and Tucker formulations. We note that recent work GRET (Chen et al., 2025) also explores temporal tensor decomposition with neural ODE components; however, we did not include it as a baseline due to unavailability of open-source implementation.

## C ADDITIONAL EXPERIMENTS

### C.1 CORESET SELECTION EFFECTIVENESS

The visualization of factor trajectories across different datasets demonstrates SONATA's capability to capture diverse temporal patterns. As shown in Fig. 5, our model effectively learns the temporal evolution of entity embeddings while quantifying uncertainty through confidence bands. This aligns with the framework's synergistic coreset strategy that dynamically selects informative data points by assessing their potential for uncertainty reduction and pattern introduction.

As shown in Fig. 6, the temporal patterns of entities selected by SONATA's coreset criteria in the server monitoring dataset reveal the highest-scoring factor with a score of 0.800 in the top panel, characterized by well-defined, periodic spikes at regular intervals with relatively low uncertainty (narrower confidence bands). This pattern likely corresponds to scheduled server activities or predictable system behaviors that SONATA correctly identifies as highly informative.

In contrast, the bottom panel displays the lowest-scoring factor with a score of 0.449, exhibiting significantly higher variability, irregular fluctuations, and wider uncertainty bands. This comparison demonstrates SONATA's ability to effectively distinguish between high-value patterns containing concentrated, reliable information and noisy patterns with less predictive value.

The clear visual difference between these factors validates our synergistic coreset criteria, which prioritizes entities based on their latent novelty, influence, and uncertainty characteristics. This selection mechanism ensures that computational resources are allocated to the most informative components of the tensor representation, leading to more efficient and accurate dynamic modeling of server performance data.

---

[2]https://sites.google.com/eng.ucsd.edu/fitrec-project/home
[3]https://zenodo.org/record/3610078#.Y8SYt3bMJGi
[4]https://archive.ics.uci.edu/ml/datasets/Beijing+Multi-Site+Air-Quality+Data

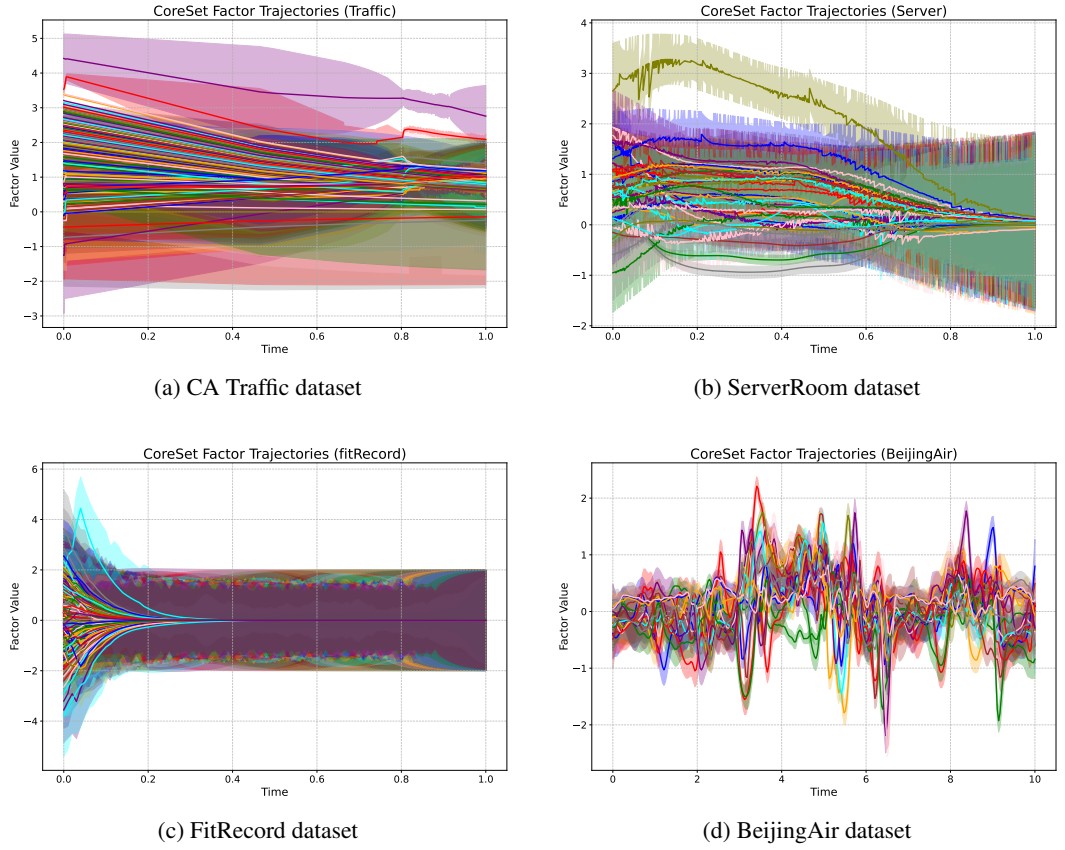

(a) CA Traffic dataset

(b) ServerRoom dataset

(c) FitRecord dataset

(d) BeijingAir dataset

Figure 5: CoreSet Factor Trajectories across different datasets.

## C.2 EMPIRICAL ANALYSIS OF CORESET MECHANISM

To comprehensively evaluate the effectiveness of SONATA's synergistic coreset strategy, we conducted extensive experiments examining three critical aspects: (1) comparison with processing all data points, (2) comparison with simple random sampling, and (3) performance across different coreset budgets.

First, we validated the computational efficiency gains of our coreset mechanism by comparing SONATA's performance when processing all available data points versus using our synergistic coreset selection. Experiments were conducted on the ServerRoom dataset with 10,000 total observations. Processing all data points increases computational cost by over 25× (from 0.338s to 8.5s per iteration) while providing negligible improvements in prediction accuracy (RMSE: 0.1290 vs 0.1293, MAE: 0.0942 vs 0.0940). This demonstrates that our coreset mechanism successfully identifies and retains the most informative samples while discarding redundant information, achieving substantial computational savings without sacrificing predictive performance. For larger datasets like CA Traffic 30K, processing every data point becomes computationally prohibitive, making efficient strategies like coresets essential.

To isolate the contribution of our synergistic selection strategy, we compared SONATA's multi-criteria coreset selection against simple random sampling using the same coreset budget (400 samples) on the ServerRoom dataset. Our synergistic coreset strategy significantly outperforms random sampling, reducing RMSE by 12.7% (0.1293 vs 0.1481) and MAE by 12.6% (0.0940 vs 0.1075). This substantial improvement validates our core claim that SONATA's performance gains stem not merely from sub-sampling, but specifically from its intelligent selection mechanism that evaluates uncertainty, influence, novelty, and information gain to capture critical events (e.g., anomalous temperature patterns preceding system failures) that random sampling might miss.

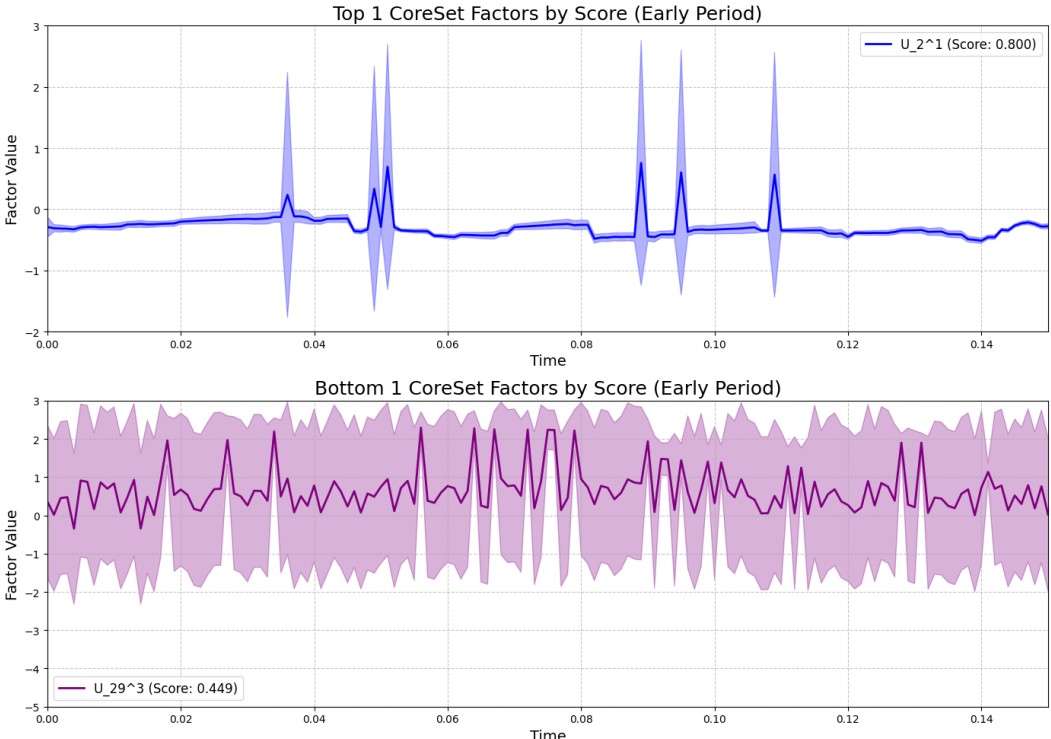

Figure 6: Visualization of SONATA's CoreSet selection effectiveness on Server Dataset. The top panel shows the highest-scoring factor ($U_2^1$, Score: 0.800) with well-defined, periodic spikes and low uncertainty bands. The bottom panel displays the lowest-scoring factor ($U_{29}^3$, Score: 0.449) exhibiting high variability and wider uncertainty bands, demonstrating SONATA's ability to effectively identify informative patterns versus noisy ones.

These empirical studies collectively demonstrate that SONATA's synergistic coreset mechanism is both computationally efficient and strategically effective, achieving near-optimal performance with only a small fraction of the total data through intelligent, multi-criteria-based selection.

### C.3    ONLINE PREDICTION ERROR

As shown in Fig. 7, we evaluate the online prediction performance on the CA traffic 30k dataset. The results demonstrate that SONATA consistently achieves lower and more stable RMSE compared to SONATA, especially in the early stages. While both methods show convergence after processing around 20,000 entries, SONATA maintains a slight advantage in prediction accuracy.

### C.4    ANALYSIS OF CORESET IMPORTANCE SCORE COMPONENT WEIGHTS

The SONATA framework utilizes a synergistic coreset selection strategy where the importance score $S_n$ for each data point is a weighted sum of four components: uncertainty reduction ($I_{unc}$), influence ($I_{inf}$), novelty ($I_{nov}$), and Martingale-based information increment ($I_{mart}$). The respective non-negative weights $w_u, w_i, w_n, w_m$ balance the contributions of these components. As shown in Table 4, we conducted experiments to evaluate the impact of different weighting schemes on the model's performance, measured by Root Mean Square Error (RMSE) and Mean Absolute Error (MAE). Additionally, we analyzed the effect of different discount factors on model performance, as presented in Table 3, and the impact of rank selection shown in Table 5.

The results presented in Table 4 indicate that the choice of weights for the different components of the importance score $S_n$ has a discernible impact on SONATA's predictive performance. Among the scenarios where only a single component was active, prioritizing "Influence" ($w_i = 1$) yielded the

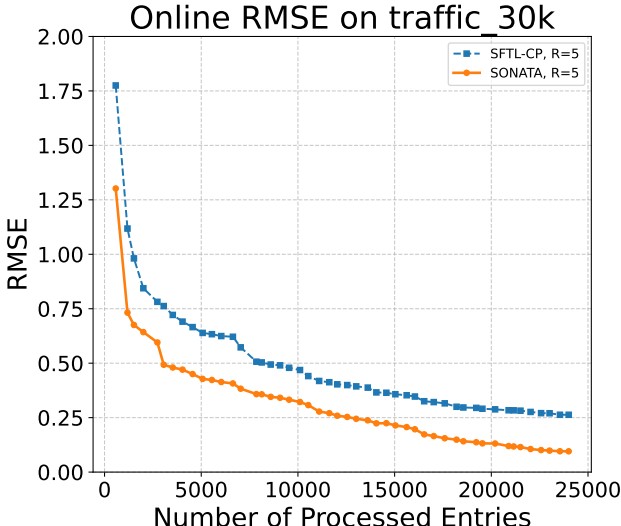

Figure 7: Online RMSE comparison between SFTL-CP and SONATA (R=5) on traffic_30k dataset.

Table 4: Effect of Coreset Importance Score Component Weights on SONATA Model Performance

| Weights | | | | Performance | |
|---|---|---|---|---|---|
| $w_u$ (Uncertainty) | $w_i$ (Influence) | $w_n$ (Novelty) | $w_m$ (Martingale) | RMSE | MAE |
| 1.00 | 0.00 | 0.00 | 0.00 | 0.1207 | 0.0866 |
| 0.00 | 1.00 | 0.00 | 0.00 | 0.1162 | 0.0845 |
| 0.00 | 0.00 | 1.00 | 0.00 | 0.1172 | 0.0842 |
| 0.00 | 0.00 | 0.00 | 1.00 | 0.1214 | 0.0888 |
| 0.00 | 0.33 | 0.33 | 0.34 | 0.1179 | 0.0851 |
| 0.33 | 0.00 | 0.33 | 0.34 | 0.1138 | 0.0820 |
| 0.33 | 0.33 | 0.00 | 0.34 | 0.1169 | 0.0842 |
| 0.34 | 0.33 | 0.33 | 0.00 | 0.1168 | 0.0840 |

lowest RMSE (0.1162) and MAE (0.0845) compared to exclusively using Uncertainty, Novelty, or the Martingale increment.

However, the best overall performance in this experiment was achieved with a combined weighting scheme. Specifically, the combination of weights $[w_u = 0.33, w_i = 0.00, w_n = 0.33, w_m = 0.34]$ resulted in the lowest RMSE of 0.1138 and the lowest MAE of 0.0820. This suggests that for the tested dataset and configuration, a strategy that balances Uncertainty, Novelty, and the Martingale information increment, while placing minimal or no emphasis on the Influence component, is most effective for constructing an informative coreset and achieving higher predictive accuracy. This demonstrates the synergistic nature of the coreset criteria, where a thoughtful combination of factors can outperform individual heuristics.

### C.5 Effect of Rank

The rank $R$ in tensor decomposition determines the number of latent factors used to represent the data. We evaluated the performance of SONATA on the Server dataset with different rank values, as shown in Table 5.

The results in Table 5 indicate that for the Server dataset, a rank of 3 achieved the lowest RMSE (0.1233) and MAE (0.0902). Increasing the rank to 5 or 7 did not lead to improved performance; in fact, the RMSE and MAE slightly increased. This suggests that a rank of 3 is sufficient to capture the dominant underlying patterns in the Server dataset, and higher ranks might introduce unnecessary

Table 5: Effect of Rank on SONATA Model Performance (Server Dataset)

| Rank | RMSE | MAE |
|------|------|-----|
| 3 | 0.1233 | 0.0902 |
| 5 | 0.1293 | 0.0940 |
| 7 | 0.1256 | 0.0915 |

complexity or lead to overfitting. The paper mentions that the main experimental results in Table 1 were obtained with $R = 5$. While $R = 3$ shows better results in this specific sensitivity analysis for the Server dataset, $R = 5$ might have been chosen as a general setting or based on performance across multiple datasets or other considerations not detailed in this snippet.

## C.6 KALMAN FILTER AND EP COMPARED

We compare the Kalman Filter and EP, and present the results and the final performance measurements, RMSE and MAE, with either method in the table below:

Table 6: Comparison of Kalman Filter and EP (SONATA) Performance

| Method | RMSE | MAE |
|--------|------|-----|
| Kalman Filter | 0.2515 | 0.1876 |
| EP (SONATA) | 0.0891 | 0.0150 |

Note that a significant RMSE decrease is possible to be made in EP with respect to Kalman filtering. This improvement was primarily due to three key factors. First, Kalman filters struggle with non-linear observations, especially the products of multiple factors in tensor CP decomposition. At this stage it can only linearize the approximation. However, in EP, nonlinear factors are naturally dealt with and thus the conclusion becomes more accurate. Thereafter, EP can share the information with all the relevant entities via message passing and thus integrate the global info. In contrast, Kalman filtering updates locally, which can reduce its performance for more complicated problems. Finally, the pseudo-observation method of EP offers excellent characteristics for tensor structures and is more accurate for such contexts. These benefits illustrate that EP is critical for the case of nonlinear tensor flow problems, which Kalman filtering is less suitable to deal with.

## C.7 SCALABILITY ANALYSIS

In this subsection, we present a scalability analysis of the proposed method on the Traffic dataset for various data sizes. Table 7, summarizes the experimental results.

Table 7: Scalability analysis on the Traffic dataset with varying data sizes.

| Data Size | Final RMSE | Peak Memory (MB) | Running Time (s) | Final Coreset Size |
|-----------|-----------|-------------------|-------------------|---------------------|
| $1,000$ | 0.5920 | 4.24 | 88.49 | 345 |
| $5,000$ | 0.4236 | 2.68 | 315.24 | 827 |
| $10,000$ | 0.2358 | 4.12 | 403.97 | $1,153$ |
| $30,000$ | 0.0891 | 8.52 | 849.83 | $1,654$ |

The results show that the proposed approach has good scalability and near-linear time complexity. With the size of the data increased from $1,000$ to $30,000$ points (i.e., $30\times$ time), the running time grows from $88.49$ seconds to $849.83$ seconds (approximately $9.6\times$ increase in time). This sublinear scaling is achieved using a simple coreset function: the more data points they process, the more the algorithm keeps those samples which are informative with respect to each other, causing a sub-linear increase in the active coreset size. The method maintains only $1,654$ of core samples, accounting for approximately $5.5\%$ of the total $30,000$ number of sets. In addition, memory usage is maintained throughout the entire process and is largely determined by coreset size rather than the entire dataset size. The results validate our method as an efficient and scalable computation solution to be used for large-scale processing data streams.

## C.8 Handling Irregular Time Steps

Table 8: Performance under different temporal sampling patterns. SONATA demonstrates robustness to irregular time steps.

| Sampling Mode | Final RMSE | Time Steps | Interval Mean | Interval Std |
|---|---|---|---|---|
| Regular | 0.0891 | 217 | 0.0046 | 0.0330 |
| Random Dropout | 0.1262 | 154 | 0.0066 | 0.0420 |
| Bursty Sampling | 0.1606 | 72 | 0.0137 | 0.0832 |
| Exponential Gaps | 0.1329 | 56 | 0.0043 | 0.0044 |

A key advantage of SONATA is that it can inherently treat irregular time steps like our continuous-time SDE formulation $dx/dt = Fx(t) + Lw(t)$. Discretized, the state transition matrix $A(\Delta t) = \exp(F \cdot \Delta t)$ naturally accommodates arbitrary time intervals $\Delta t$ between observations. To systematically analyze this property, we performed experiments in different irregular sampling conditions and we could find results in Table 8. Our methodology proved to be robust as confirmed by the results. SONATA performs best (RMSE = 0.0891) under normal sampling. Performance declines gracefully if your data has missing observations (*Random Dropout*) or highly irregular patterns (*Bursty Sampling* and *Exponential Gaps*). Nevertheless even in the *Bursty Sampling* hard scenario, with high variance in time spans (Std = 0.0832), SONATA has good reproducibility (RMSE = 0.1606). This illustrates the real-world applicability of our continuous-time method for deployment and actual real-time application situations where data acquisition is irregular and sometimes unpredictable.

## D LLM Usage Disclosure.

In accordance with ICLR 2026 policy, we disclose our use of Large Language Models (LLMs) in the preparation of this work. We employed LLM tools, including OpenAI's GPT series, to assist in polishing the writing of this manuscript. Their role was limited to improving clarity, grammar, and readability of certain sections such as the Introduction and Related Work, as well as helping rephrase sentences for stylistic consistency. During the research process, we occasionally used LLMs for brainstorming alternative formulations of background text and related work discussions, but all technical content, theoretical analysis, algorithmic design, and empirical experiments were conceived and executed entirely by the authors. We used LLMs for programming support including code formatting, commenting, and implementation assistance, but the core design of the SONATA framework, the coreset mechanism, Bayesian inference pipeline, and all experimental evaluations were independently developed and validated by the research team. All LLM-generated suggestions were carefully reviewed and revised by the authors. The core scientific contributions including problem formulation, algorithmic innovations, proofs, and experimental analyses remain completely original and the sole work of the authors.

