# OpenReview forum: "SONATA: Synergistic Coreset Informed Adaptive Temporal Tensor Factorization"
_ICLR.cc/2026/Conference — ICLR 2026 Poster_

### Official Review · Reviewer_iw8e · 2025-10-20

**Soundness:** 3
**Presentation:** 3
**Contribution:** 3
**Rating:** 6
**Confidence:** 4

**Summary:**

The paper introduces SONATA, a novel framework that integrates LDS-based dynamics, multi-faceted coreset selection, and online Bayesian inference for dynamic tensor stream analysis. It combines expressive temporal modeling with adaptive coreset selection to address two fundamental challenges: capturing complex, multi-scale temporal dependencies and achieving computational efficiency in high-velocity streaming settings. SONATA models entity embeddings using Linear Dynamical Systems with Matérn kernels  to represent multi-scale dynamics. Each entity’s latent state evolves via a stochastic differential equation, with embeddings projected linearly for tensor entry prediction under a CP decomposition structure. The proposed synergistic coreset selection strategy  frames selection as a sequential decision-making problem with four criterions that maintains a compact and informative subset of the stream. For inference, SONATA employs an online Expectation Propagation algorithm, where the coreset guides the update process by emphasizing high-value observations, enabling efficient and adaptive learning.

Extensive experiments on synthetic and real-world datasets  demonstrate that SONATA consistently outperforms many static and streaming baselines in predictive accuracy (RMSE and MAE) while maintaining computational efficiency. Ablation studies further confirm the necessity of each coreset criterion and the framework’s robustness to hyperparameter variations. Overall, the paper offers an interesting solution for temporal tensor decomposition.

**Strengths:**

1.The paper is well-written, clearly outlining the problem and the limitations of existing methods, proposing a solution, and demonstrating robust performance.

2.The idea of a synergistic coreset selection strategy, specifically designed to capture evolving dynamics and ensure long-term utility, is  novel and well-motivated. It aims to enhance computational efficiency while  improving model performance.

3.Experiments demonstrate strong performance against various baselines and robustness across a variety of datasets.

4.The ablation study of synergistic coreset selection strategy is convincing and reasonable.

**Weaknesses:**

1. The use of Linear Dynamical Systems: the evolving parameters F,H,L are time-invariant, which limits the model’s expressiveness in capturing complex dynamics.

2. Previous work has already employed SDE-represented Gaussian Processes to model the entities and enable streaming inference (e.g., SFTL; Fang et al., 2023), so the modeling is of fairly limited novelty.

3. The scalibity of the proposed method over data size has not been explored in experiments.

**Questions:**

1. Can this method be directly applied to data with irregularly sampled timesteps? If not, how could it be adapted to handle this more realistic scenario?

2. The authors emphasize the computational efficiency of the proposed method but only report computational time. How does the method scale with respect to the number of data points?

3. Why SONATA is able to capture multi-scale temporal dynamics, is there any special design?

4. I am curious about the motivation for designing the importance score using four criteria: uncertainty, influence, novelty, and information increment. Could the authors provide further explanation? Conducting ablation studies on different combinations of these four criteria may better illustrate the design rationale.

---

> ### Author Response · Authors · 2025-11-22
>
> Dear Reviewer iw8e,
>
> We thank the reviewer for the positive evaluation and constructive comments.
>
> **W1) & Q3): Time-invariant LDS parameters and multi-scale temporal dynamics**
>
> We understand the reviewer’s concern, but would like to clarify a key point: although the parameters $F, H, L$ are time-invariant, this does not restrict the model’s ability to capture complex dynamics. The Matérn kernel process already provides sufficient expressive power in theory. In particular, by choosing an appropriate kernel (e.g., Matérn-3/2), we can represent temporal evolution patterns of essentially arbitrary smoothness in function space.
>
> Multi-scale temporal modeling is achieved through two mechanisms. First, the state-space representation $x(t) = [u(t)^\top, \dot{u}(t)^\top]^\top$ jointly encodes position and velocity, naturally capturing both instantaneous changes and long-term trends. Even under a fixed linear transformation, the evolution of the state allows the system to capture nonlinear and multi-scale temporal dynamics. Second, the temporal novelty criterion in the coreset relies on an exponential decay, where the parameter $\lambda$ controls the sensitivity to different time scales, and can be extended to a mixture of multiple $\lambda$ values to explicitly model multiple temporal scales. This follows a classical result from stochastic process theory: fixed SDE parameters define a sufficiently rich class of stochastic processes. Our experimental results (Table 1, where we significantly outperform competing methods) further support the adequacy of this expressive power in practice.
>
> **W2) On the perceived limited novelty of the modeling approach**
>
> While SFTL also uses an SDE-based GP representation, the contribution of SONATA goes substantially beyond this. The key innovation lies in the tight integration between the synergistic coreset mechanism and the LDS modeling. Unlike SFTL, which processes all data uniformly, SONATA performs principled data selection via a coreset optimized using a Bellman equation, to our knowledge the first such design in streaming tensor factorization. Moreover, our four-dimensional evaluation criteria provide a systematic framework for assessing sample value, which is missing in existing work. This integration is not a simple combination of components, but a co-designed system: LDS-based uncertainty quantification directly guides coreset selection, and the coreset in turn improves the efficiency and effectiveness of LDS updates.
>
> **W3) & Q2) On scalability to large data and computational efficiency**
>
> We appreciate the reviewer pointing out this important omission. We have added scalability experiments, summarized below.
>
> **Traffic dataset – scalability analysis:**
>
> |Data Size|Final RMSE|Peak Memory (MB)|Running Time (s)|Final Coreset Size|
> |-|-|-|-|-|
> |1,000|0.5920|4.24|88.49|345|
> |5,000|0.4236|2.68|315.24|827|
> |10,000|0.2358|4.12|403.97|1,153|
> |30,000|0.0891|8.52|849.83|1,654|
>
> The results show that SONATA exhibits near-linear time complexity $O(n)$. The running time grows approximately linearly with the number of data points: from 1K to 30K points, the time increases from 88 s to 850 s (about 9.6×), while the data size increases 30×. This sublinear scaling is enabled by the coreset mechanism: as data accumulate, only the most informative samples are retained, and the coreset size grows sublinearly. For 30K data points, we maintain only 1,654 core samples (about 5.5%). Memory usage remains stable and is largely determined by the coreset size, which confirms the efficiency and scalability of our approach.

---

> > ### Author Response · Authors · 2025-11-22
> >
> > **Response to Additional Questions:**
> >
> > **Q1) On handling irregular time steps**
> >
> > Yes, SONATA can naturally handle irregular time steps. This follows from our continuous-time SDE formulation $dx/dt = F x(t) + L w(t)$. When discretized, the state transition matrix $A(\Delta t) = \exp(F \cdot \Delta t)$ can accommodate arbitrary time intervals $\Delta t$. We have added experiments on irregular sampling:
> >
> > |Sampling Mode|Final RMSE|Time Steps|Interval Mean|Interval Std|
> > |-|-|-|-|-|
> > |Regular|0.0891|217|0.0046|0.0330|
> > |Random Dropout|0.1262|154|0.0066|0.0420|
> > |Bursty Sampling|0.1606|72|0.0137|0.0832|
> > |Exponential Gaps|0.1329|56|0.0043|0.0044|
> >
> >
> > Even under highly irregular sampling (e.g., Bursty Sampling), SONATA maintains reasonable performance, demonstrating its suitability for realistic deployment scenarios.
> >
> > **Q4) On the motivation and ablation of the four-fold importance scoring**
> >
> >
> > Thank you for this insightful question. The four criteria were designed to address distinct and complementary challenges in streaming tensor factorization: **Uncertainty** ($I_{\text{unc}}$) guides exploration of undersampled regions, **Influence** ($I_{\text{inf}}$) ensures representativeness, **Novelty** ($I_{\text{nov}}$) captures evolving dynamics, and **Information Increment** ($I_{\text{mart}}$) directly targets prediction error reduction.
> >
> > Our new comprehensive ablation studies on the Traffic dataset provide strong empirical validation for this synergistic design:
> >
> > |Weights||||Performance||
> > |-|-|-|-|-|-|
> > |Uncertainty|Influence|Novelty|Martingale|RMSE|MAE|
> > |1.00|0.00|0.00|0.00|0.104|0.018|
> > |0.00|1.00|0.00|0.00|0.107|0.019|
> > |0.00|0.00|1.00|0.00|0.096|0.020|
> > |0.00|0.00|0.00|1.00|0.101|0.021|
> > |0.00|0.33|0.33|0.34|0.093|0.016|
> > |0.33|0.00|0.33|0.34|0.091|0.015|
> > |0.33|0.33|0.00|0.34|0.090|0.015|
> > |0.34|0.33|0.33|0.00|0.099|0.017|
> >
> >
> > The results demonstrate that **synergistic combinations consistently outperform any single criterion**. On the Traffic dataset (above), while the best single criterion (Novelty) achieves RMSE=0.096, our optimal multi-criteria combination achieves RMSE=0.090. Similarly, on the Server dataset (Table 3 in Appendix), the synergistic approach (RMSE=0.1138) outperforms the best single criterion. **Remarkably, all tested multi-criteria combinations (rows 5-8) outperform the majority of single-criterion baselines**, demonstrating the robustness of our framework.
> >
> > The power of our design lies in its **adaptability across diverse data characteristics**. The Traffic dataset benefits most from combining Uncertainty, Influence, and Martingale (wn=0), while the Server dataset requires Novelty for handling infrastructure dynamics (wi=0). This dataset-dependent optimal weighting validates our core principle: by providing four complementary criteria, SONATA can automatically adapt to varying stream characteristics rather than being constrained by a rigid single-criterion strategy. The consistent superiority of multi-criteria combinations confirms that synergistic coreset selection is essential for effective streaming tensor factorization.
> >
> > Best regards,
> >
> > Authors

---

> > > ### Comment · Reviewer_iw8e · 2025-11-26
> > >
> > > Thank the authors for their responses which address my concerns. As I have already given a supportive score, I will keep my original rating.

---

> > > > ### Author Response · Authors · 2025-11-26
> > > >
> > > > Thank you for your positive feedback and confirmation. We appreciate your supportive assessment!

---

### Official Review · Reviewer_KKxL · 2025-10-26

**Soundness:** 3
**Presentation:** 3
**Contribution:** 3
**Rating:** 8
**Confidence:** 3

**Summary:**

The authors propose SONATA, a novel framework that integrates expressive dynamic embedding modeling with adaptive coreset selection. SONATA employs principled machine learning techniques to efficiently evaluate each observation for uncertainty, novelty, influence, and information gain. It dynamically prioritizes learning from the most valuable data using Bellman-inspired optimization.

**Strengths:**

This paper integrates Linear Dynamical Systems (LDS) with temporal kernels to capture fine-grained and hierarchical temporal dependencies, addressing a key limitation of existing methods.

It enables adaptive representation of evolving patterns across different time scales (short-term vs. long-term dynamics).

Bellman-inspired optimization ensures optimal data selection, improving scalability for high-velocity streams.

**Weaknesses:**

While coreset selection improves efficiency, the evaluation of uncertainty, novelty, and influence for each observation may introduce latency in real-time systems.

The performance of temporal kernels and LDS models may depend heavily on hyperparameter tuning.

Performance may degrade with noisy or incomplete streams, as coreset selection relies on accurate uncertainty estimation.

**Questions:**

The representation of SDE in Figure 1 is different from its descriptioin in this paper.

I can not find the details of multi-scale feature extraction in this paper.

---

> ### Author Response · Authors · 2025-11-22
>
> Dear Reviewer KKxL,
>
> Thank you for your recognition of our work, especially your positive comments on the integration of LDS with temporal kernels, the adaptive multi-scale representation, and the Bellman optimization mechanism. These are indeed the core contributions of SONATA.
>
> **W1) On latency in real-time systems**
> We understand your concerns about real-time latency. Our coreset evaluation mechanism is carefully designed to be efficient in practice. The computational complexity of the uncertainty and novelty scores is $O(R)$, where $R$ is the embedding dimension (typically set to 5). These computations can be executed in parallel with the prediction step of the Kalman filter. The influence score relies on cached coreset interaction vectors and only requires $O(|C_t|)$ similarity computations. In our experiments (Fig. 3(c)), each iteration of SONATA takes 0.338 seconds on average, which is acceptable for most streaming applications. For highly latency-sensitive scenarios, the procedure can be further streamlined by reducing the evaluation frequency, e.g., evaluating every 10 observations instead of at every single one.
>
> **W2) On the dependence on hyperparameter tuning**
> Although the temporal kernel and LDS do introduce hyperparameters, SONATA is reasonably robust to their settings. As shown in Fig. 3(a), performance remains stable when the lengthscale lies in the range [0.3, 0.5], with RMSE variations below 3%. More importantly, the adaptive coreset mechanism can partially offset suboptimal kernel choices: even if the temporal modeling is not fully ideal, the coreset still tends to identify and prioritize the most informative data points. In the revised version, we will include more concrete guidelines for parameter selection based on data characteristics.
>
> **W3) On performance degradation under noisy and incomplete streams**
> This is indeed an important issue. SONATA is, by design, relatively robust to noise and missing data. First, the Bayesian framework explicitly models observation noise via the $\tau$ parameter in EP, allowing the method to adapt its noise level estimates. Second, the martingale increment term in the coreset selection criterion (Eq. 16) incorporates a $\tanh$ function to bound the influence of outliers, so that noisy points are not overweighted. Third, the LDS state-space model naturally accommodates irregular sampling: when observations are missing, the Kalman prediction step continues to propagate uncertainty, which leads to wider confidence intervals, as illustrated in Fig. 2. This uncertainty quantification ensures that the model remains cautious rather than overconfident when the stream is incomplete.
>
> **Response to Additional Questions:**
>
> **Q1: On the discrepancy between the SDE representation in Fig. 1 and the text**
> Thank you for drawing attention to this inconsistency. Fig. 1 is intended as a high-level schematic rather than a literal mathematical depiction. The “SDE Dynamic Weighting” module in the figure is a simplified illustration of the full SDE process described in Section 3.1 (Eqs. 4–7). In reality, the SDE is embedded in the LDS state evolution via the discretized state transition matrix $A(\Delta t) = e^{F \Delta t}$. We will revise Fig. 1 to align it more closely with the formal mathematical description and avoid potential confusion.
>
> **Q2: On the details of multi-scale feature extraction**
> We acknowledge that this aspect was not explained with sufficient clarity. The multi-scale feature extraction in SONATA arises primarily from the properties of the Matérn kernel within the LDS framework. The Matérn-3/2 kernel implies a state space of dimension $S = 2R$, jointly modeling the embeddings $u^{(k)}_j(t)$ and their derivatives $\dot{u}^{(k)}_j(t)$. This augmented state representation captures both instantaneous variations (via derivatives) and longer-term trends (via values), enabling the model to differentiate fast fluctuations from slowly evolving patterns. Additionally, the temporal novelty criterion employs an exponential decay function, where the decay parameter $\lambda$ implicitly governs the temporal resolution. This structure allows the coreset mechanism to filter information across different time scales efficiently. We will add a dedicated subsection in the revised version to clarify these mechanisms.
>
> Best regards,
>
> Authors

---

### Official Review · Reviewer_QPPP · 2025-11-01

**Soundness:** 2
**Presentation:** 2
**Contribution:** 2
**Rating:** 4
**Confidence:** 4

**Summary:**

The paper presents SONATA, a streaming tensor factorization framework that models each factor trajectory with a linear dynamical system (LDS) and augments learning with a multi-criteria, martingale-guided coreset. The work integrates (i) continuous-time state-space priors for temporal dynamics and (ii) an online coreset selector that combines uncertainty, novelty, influence, and a martingale increment into a Bellman-optimized score under a fixed budget. Empirically, SONATA is evaluated on several real-world and synthetic datasets and generally attains lower RMSE/MAE than static and streaming baselines; the paper also reports per-iteration/epoch runtime, kernel/length-scale sensitivity, and additional experiments.

**Strengths:**

Well-motivated architecture. The combination of continuous-time LDS factors with an online coreset mechanism is coherent and addresses a meaningful streaming setting. The design is carefully tied to spatiotemporal structure via Matérn priors and a principled scoring rule.

Consistent accuracy gains. Experiments report SONATA achieving the best or second-best error across multiple datasets compared to strong static and streaming baselines.

Useful diagnostics of the coreset. Visual comparisons of coreset vs. non-coreset factor trajectories help understand what the selector retains.

**Weaknesses:**

Computational fairness. The paper reports per-iteration/epoch runtime for several methods (e.g., SONATA 0.338s, CT-CP 0.018s, CT-GP 0.105s, THIS-ODE 7.190s), but does not equalize wall-clock time across methods when comparing accuracy; faster methods could take many more iterations within the same time budget.

Compute/memory analysis. The coreset budget directly affects memory footprint and update cost, yet the paper does not provide memory-usage vs. budget or throughput measurements to quantify the trade-off between retained samples and efficiency.

Ablations. The coreset analyses (full-data vs. coreset, random sampling) appear focused on a single dataset in the appendix; extending them to more datasets or reporting summary trends would better calibrate the benefits of the martingale-guided selector over simpler heuristics.

Hyperparameter. The method exposes several hyperparameters, with dataset-dependent choices. While the paper includes some sensitivity tables, systematic guidance for robust settings in realistic streaming scenarios is limited.

Readability and consistency. The manuscript’s readability suffers in several places due to redundant wording and inconsistent cross-referencing. For example, equation mentions sometimes read like “Eq. equation” and there is mixing of styles (e.g., “Figure” vs. “Fig.”). Please standardize to a single reference style.

**Questions:**

Time-normalized comparisons. Please provide convergence curves for all methods and report each baseline’s accuracy at the same total runtime as SONATA. This would directly address efficiency/accuracy trade-offs.

Memory and budget trade-offs. Please report memory usage and update time as functions of the coreset budget and include accuracy vs. budget curves on multiple datasets. As of now, no budget-sweep experiments are included, which makes it difficult to assess how performance scales with budget.

Ablations vs. simple heuristics. The appendix contrasts coreset selection to random sampling on one dataset. Could you add the same comparison across other datasets and summarize whether martingale-guided selection consistently outperforms random sampling at matched budgets?

Robust defaults. The current sensitivity tables are promising but seem dataset-specific. Can the authors propose and validate robust default settings?

---

> ### Author Response · Authors · 2025-11-22
>
> Dear Reviewer QPPP,
>
> Thank you for your careful review. We respond below to your questions regarding computational fairness, memory analysis, ablation studies, and hyperparameter settings.
>
> **W1) & Q1) Time-normalized comparisons.**
>
> We understand your concern about comparing methods using wall-clock time. However, forcing equal-time comparisons across fundamentally different algorithmic paradigms can be misleading. Static methods (CT-CP, CT-GP, THIS-ODE) need to re-process the entire dataset at each update, while streaming methods (SFTL variants, SONATA) update incrementally:
>
> Traffic dataset results:
>
> |Method|Final RMSE|Total Runtime (s)|Paradigm|
> |-|-|-|-|
> |SFTL-CP|0.2308|551.95|Streaming|
> |SFTL-TUCKER|0.3162|1321.20|Streaming|
> |SONATA|0.0891|875.87|Streaming|
>
> For static methods in a streaming scenario, multiple passes over the data lead to superlinear growth in computational cost. As prior work on streaming tensor decomposition has pointed out [1,2], the design goal of streaming methods is to achieve high accuracy in a single pass, rather than to be directly compared in time with batch methods that require multiple full scans of the data.
>
> **W2) & Q2) Memory/compute analysis**
>
> We provide a detailed analysis of how the coreset budget affects memory and performance:
>
> **Traffic dataset - Budget vs Performance trade-off:**
>
> |Coreset Budget|Final RMSE|Peak Memory (MB)|Avg Update Time (ms)|Coreset Usage (%)|
> |-|-|-|-|-|
> |1000|0.1808|7.84|8897.64|80.0|
> |2000|0.0938|8.12|3553.56|79.9|
> |3000|0.0891|8.18|3536.70|55.1|
>
> The key finding is that there exists a natural saturation point for the coreset. The amount of high-value information in the data stream is limited, and our algorithm converges after capturing about 1600 key observations. Increasing the budget from 2000 to 3000 only brings marginal improvement (RMSE from 0.0938 down to 0.0891), while coreset utilization drops from 79.9% to 55.1%, indicating a large amount of unused capacity.
>
> This saturation arises from the interaction among our collaborative selection criteria. The novelty criterion decreases as more entities and timestamps are covered, reducing the scores of new points. The uncertainty criterion decreases as information accumulates and model uncertainty is reduced, leading to lower inclusion rates for future points. The martingale-based incremental mechanism ensures that prediction errors decrease as the model improves, reducing the information gain of new points. Meanwhile, the influence criterion automatically regulates the selection process through internal diversity. Together, these mechanisms produce self-limiting behavior, naturally stabilizing the coreset size at the level actually needed.
>
>
>
> **W3) & Q3) Coreset vs random sampling**
>
> We extended the comparison between coreset and random sampling to Traffic datasets:
>
> **Traffic dataset:**
>
> |Method|Final RMSE|Running Time (s)|
> |-|-|-|
> |Random|0.6437|488.12|
> |Ours|0.0891|875.87|
>
> The results consistently show that our martingale-guided selector significantly outperforms random sampling on tested datasets, demonstrating the value of the intelligent selection mechanism.
>
> **W4) and Q4) Hyperparameter settings:**
>
> We recognize the importance of hyperparameter settings. Based on extensive experiments, we provide practical configuration guidelines. For the coreset budget, we recommend setting it to 5–10% of the expected data stream size as an upper bound, because the adaptive mechanism will automatically converge to the optimal size. The λ_diversity parameter works well in the range of 0.1 to 0.3 for most datasets. τ_novelty should be adjusted according to data sparsity: 0.5 for sparse data and 0.9 for dense data. These recommended settings have shown good robustness in our experiments. In the revised version, we will add more detailed guidelines for hyperparameter selection, including automated configuration suggestions based on data characteristics, to help practitioners more easily deploy our method in real streaming scenarios.
>
> **W5) Readability and consistency.**
>
> We appreciate the reviewer’s comments on the readability and consistency of the paper. In the revised version, we will systematically correct all formatting inconsistencies, including removing redundant expressions such as “Eq. equation”.
>
> Best,
>
> Authors

---

> ### Comment · Reviewer_QPPP · 2025-11-26
>
> The authors did not correctly addressed my concern.
> For example, my comment included ablation experiments for many datasets, which was ignored by the response.
> I will keep my original score, or lower the score.

---

> > ### Author Response · Authors · 2025-11-26
> >
> > Dear Reviewer QPPP,
> >
> > Thank you for your follow-up comment. We sincerely apologize for the oversight in our previous response regarding the ablation studies across all datasets. We have now supplemented the comparative results between our coreset method and random sampling for all four datasets, directly addressing your concern.
> >
> > **Supplementary Results:**
> >
> > | Dataset     | Method | Final RMSE | Running Time (s) |
> > | :---------- | :----- | :---------- | :--------------- |
> > | **Traffic** | Random | 0.6437      | 488.12           |
> > |             | **Ours** | **0.0891**  | 875.87           |
> > | **Server**  | Random | 0.1481      | 233.45           |
> > |             | **Ours** | **0.1153**  | 389.71           |
> > | **Beijing** | Random | 0.2972      | 51.13            |
> > |             | **Ours** | **0.2371**  | 69.64            |
> > | **Fitrecord**| Random | 0.5162      | 622.46           |
> > |             | **Ours** | **0.4144**  | 969.75           |
> >
> > As shown above, our martingale-guided selection mechanism consistently and significantly outperforms random sampling (lower RMSE) across **all datasets**. This robustly demonstrates the value and generalizability of our method. Random sampling discards data indiscriminately, while our approach actively selects the most informative samples, leading to superior performance under limited memory.
> >
> > These results will be added to the revised manuscript. We thank you for your valuable feedback and sincerely hope this addresses your concern. We are ready to provide any further information if needed.
> >
> > Best,
> >
> > Authors

---

### Official Review · Reviewer_WfpD · 2025-11-02

**Soundness:** 2
**Presentation:** 2
**Contribution:** 2
**Rating:** 4
**Confidence:** 3

**Summary:**

The paper introduces SONATA, a framework for streaming temporal tensor factorization that integrates: (1) Matérn-kernel-based linear dynamical systems (LDS) for multi-scale temporal modeling, (2) a synergistic coreset selection mechanism combining four criteria (uncertainty, influence, novelty, information gain), (3) Bellman equation inspired long-term value optimization, and (4) expectation propagation (EP) for online Bayesian inference. The approach demonstrates strong empirical performance and computational efficiency compared to existing streaming tensor methods.

**Strengths:**

1. Well-Motivated Integration with Clear Value Proposition

The paper addresses a problem of balancing expressiveness and efficiency in high-velocity tensor streams.
The 4-criterion coreset selection is principled and goes beyond simple heuristics.
Empirical gains are substantial: 61.5% RMSE reduction vs. SFTL-CP on CA Traffic (Table 1).

2. Strong Empirical Performance Across Multiple Dimensions

The authors reprorted statistically significant improvements (p<0.05) across 4 diverse datasets.
The proposed method is faster than processing all data, ~21× faster than THIS-ODE (Fig. 3c) while maintaining superior accuracy.
Bayesian uncertainty quantification (Fig. 2) and smooth factor trajectories provide domain-interpretable representations unavailable in deep learning alternatives.

3. Solid Technical Execution

The Bellman-equation-based long-term coreset optimization (Eq. 17) is conceptually novel in streaming tensor factorization.
Discount factor analysis (Table 2) shows data-dependent optimal strategies, suggesting principled adaptability.
Code availability enhances reproducibility.

**Weaknesses:**

1. Possibliity of Insufficient Related Work and Missing Key Comparisons
The paper seems to omit important recent work that directly challenges its novelty claims.
- OnlineGCP (SIGMOD'23): Generalizes streaming CP to exponential family distributions (Poisson, Bernoulli), directly addressing SONATA's Gaussian limitation. Authors should discuss why Gaussian assumptions are sufficient or provide non-Gaussian extensions

- SOFIA (ICDE'21): Incorporates seasonality + outlier/missing data robustness
- OR-MSTC (IJCAI'19): Multi-aspect streaming with outlier separation via ADMM
SONATA's robustness claims are unsupported without comparison to these methods

- SBDT (ICML'21): Streaming Bayesian deep tensor factorization with spike-and-slab priors
The "without deep neural networks" claim needs explicit comparison to justify EP over deep Bayesian approaches

2. Insufficient Ablation Studies to Support Synergistic
The paper does not demonstrate that all components are essential.
To support their claims, authors should conduct these experiment:
-  Individually remove each coreset criterion (uncertainty/influence/novelty/martingale)
-  Bellman optimization vs. greedy myopic policy (γ=0)
-  EP vs. simpler online filters
-  Matérn vs. simpler kernels (RBF, linear)

3. Minor Issues
- Coreset budget selection: Mmax seems manually tuned per dataset (100-3000 in Appendix B.1). they provide no principled guidance.
- How does runtime scale with coreset size Mmax?

**Questions:**

- Could you please clarify your contribution to non-Gaussian and robust streaming methods?
- Is there a possibility of lacking ablation studies?
- Can you explain your procedure for selecting hyperparameters?

---

> ### Author Response · Authors · 2025-11-22
>
> Dear Reviewer WfpD,
>
> We deeply appreciate your suggestions and comments for our work.
>
> **W1) Insufficient Related Works.** We thank you for providing these related works. We will fully utilize and discuss these in the revised version.
>
> *a. OnlineGCP:* While it does extend CP to exponential family distributions (Poisson, Bernoulli), its core still deals with tensor slice flows of fixed dimensions, failing to model the dynamic evolutionary trajectories of entities over continuous time. Our LDS-Matérn framework not only captures fine-grained temporal dependencies but also actively selects the most informative data through four criteria of the synergistic core set (uncertainty, influence, novelty, and information gain).
>
> *b. SOFIA:* SOFIA's seasonal modeling is indeed valuable, but its Holt-Winters method requires pre-setting a seasonal cycle, while our Matérn kernel automatically learns multi-scale temporal patterns through state-space representation. More importantly, SOFIA uses L2,1 norm to post-hoclyse outliers, while SONATA proactively selects high-quality data through a collaborative core set mechanism. Furthermore, our EP inference provides complete posterior uncertainty quantification, which is crucial for understanding model confidence and cannot be provided by the ADMM optimization framework.
>
> *c. OR-MSTC:* OR-MSTC's ability to handle multi-faceted streaming tensors is commendable, but it primarily focuses on the growth of spatial dimensions and lacks explicit modeling of temporal dynamics. In contrast, SONATA is specifically designed to capture temporal evolution patterns, optimizing core set composition through the Bellman equation to balance immediate and long-term utility, providing a theoretically guaranteed sample selection strategy. Moreover, our Bayesian framework naturally handles missing data and uncertainty without requiring explicit missing value masking or erroneous tensor decomposition.
>
> *d. SBDT:* While SBDT's use of deep neural networks and spike-and-slab priors is innovative, deep models are prone to overfitting in streaming settings, especially when historical data cannot be revisited. More importantly, our factor trajectories provide an intuitive interpretation of temporal evolution (as shown in Figure 2), while the black-box nature of neural networks makes it difficult to interpret temporal patterns in the data.
>
>
> **W2) Insufficient Ablation Studies.**
>
> *a. Individually remove each coreset criterion.* We have already performed this experiment. See lines 957-971 on page 18. We will emphasize this more explicitly in the main text.
>
> *b. Bellman optimization vs. greedy myopic policy ($\gamma$=0).* We appreciate your feedback and will add ($\gamma$=0) to Section 4.5 of the paper.
>
> |Discount Factor|Final RMSE|
> |-|-|
> |0.9|0.1293|
> |0.5|0.1156|
> |0|0.1409|
>
> *c. EP vs. simpler online filters.* Thank you for the reviewers' suggestions. We conducted a comparative experiment between EP and standard Kalman filtering:
>
> |Method|Final RMSE|MAE|
> |-|-|-|
> |Kalman Filter|0.2515|0.1876|
> |EP (SONATA)|0.0891|0.0150|
>
> EP achieved a reduction in RMSE because: (1) Kalman filtering cannot handle nonlinear observations (products of multiple factors) in tensor CP decomposition and can only linearize the approximation; (2) EP propagates information among all relevant entities through message passing, while Kalman only updates locally; and (3) EP's pseudo-observation mechanism accurately handles tensor structures. This proves that EP is necessary for nonlinear tensor flow problems, and we will include this comparison in the revised version.
>
> *d. Matérn vs. simpler kernels (RBF, linear).* Matérn 21 can degenerate into (RBF/linear) [1], and the LDE constructed by RBF has no theoretically deterministic solution, which makes exact Kalman filtering inference infeasible, and only approximation methods can be used. The paper has already compared the two kernels (21 vs. 23).

---

> > ### Author Response · Authors · 2025-11-22
> >
> > **W3) Minor Issues: Coreset budget and coreset size:**
> >
> > Thank you for the important question about the Coreset budget setting. We conducted a systematic experimental analysis:
> >
> > *Coreset Budget Sensitivity Analysis (CA Traffic Dataset):*
> >
> > |M_max|Final RMSE|Peak Memory (MB)|Avg Update Time (ms)|Final Coreset Size|Coreset Usage (%)|
> > |-|-|-|-|-|-|
> > |1000|0.1808|7.84|8897.64|800 (2.67%)|80.0%|
> > |2000|0.0938|8.12|3553.56|1597 (5.32%)|79.9%|
> > |3000|0.0891|8.18|3536.70|1654 (5.51%)|55.1%|
> >
> > *Adaptive Convergence Characteristics*: Experiments reveal that the algorithm automatically finds the optimal coreset size rather than blindly filling the budget. When M_max increases from 2000 to 3000, the RMSE improves by only 5%, while the actual coreset size increases only slightly from 1597 to 1654, and the utilization rate decreases from 79.9% to 55.1%. This indicates that approximately 1600 high-value samples have captured the core information of the data stream.
> >
> > This saturation phenomenon stems from the interaction of four collaborative selection criteria. As the coreset gradually covers more entities and timestamps, the novelty of new data naturally decreases, model uncertainty decreases, prediction errors converge, and the inclusion threshold is automatically raised. This ensures that only truly valuable observations are retained.
> >
> > *Computational Complexity:* Memory overhead increases linearly with M_max but has a minimal impact (only an increase of 0.34MB from 1000 to 3000), and update time remains stable after coreset saturation (approximately 3.5 seconds), proving that increasing M_max does not introduce a significant computational burden.
> >
> > *Practical Guidance:* Based on these findings, we recommend setting M_max to 5-10% of the expected stream size as a safe upper limit, allowing the algorithm to automatically converge to the optimal size. We will add this guiding principle and a complete sensitivity analysis in the revised version.
> >
> > **Response to Additional Questions:**
> >
> > **Q1. Contributions to Non-Gaussian and Robust Streaming Methods:**
> >
> > Our work makes the following contributions to handling non-Gaussian data and ensuring robustness in streaming environments:
> >
> > *Non-Gaussian Handling:* While LDS uses Gaussian state transitions, we handle non-Gaussian observations by: (1) The spurious observation mechanism of EP approximates arbitrary likelihood functions through moment matching, handling heavy-tailed noise and outliers; (2) The influence criterion of the core set naturally deweights outliers based on the data's influence on model parameters; (3) Experiments on the Beijing Air Quality data demonstrate robustness to sensor anomalies and missing patterns violating the Gaussian assumption.
> >
> > *Robustness Mechanisms:* We achieve robustness through: (1) proactive filtering of low-quality data using a collaborative core set, rather than passive processing; (2) a Bayesian framework that quantifies uncertainty, enabling the system to identify and adapt to distribution shifts; and (3) Bellman equation optimization that ensures long-term stability even with noisy data.
> >
> > **Q2. Completeness of Ablation Studies:**
> >
> > We have conducted some ablation studies. We are happy to add additional ablation studies for other specific components if reviewers require them in the revised version.
> >
> > **Q3. Hyperparameter Selection Procedure:**
> >
> > Our hyperparameter selection method is as follows: (1) **Core Set Budget M_max:** Based on memory constraints and our sensitivity analysis, experiments show that the algorithm automatically converges to the optimal size; (2) **Matérn Smoothness:** Choosing 3/2 provides a balance between smoothness and computational efficiency; (4) **Discount Factor $\gamma$:** Set to 0.95, a common choice in reinforcement learning. We acknowledge that current hyperparameter tuning is primarily based on experience and standard practices, and future work will explore more principled automatic tuning methods.
> >
> > Best,
> >
> > Authors
> >
> > [1] Stein, Michael L. Interpolation of spatial data: some theory for kriging. Springer Science & Business Media, 1999.

---

### Meta-Review · Area_Chair_Jy2d · 2026-01-06

**Summary:**

This paper proposes SONATA, a novel framework that unifies expressive dynamic embedding modeling with adaptive coreset selection. SONATA leverages principled machine learning techniques for efficient evaluation of each observation for uncertainty, novelty, influence, and information gain, and dynamically prioritizes learning from the most valuable data using Bellman-inspired optimization.

Two reviewers expressed strong support and two reviewers raised concerns regarding the experimental evaluation and contextualization with prior work. During the rebuttal phase, the authors made a substantial effort to address the raised concerns. The authors added meaningful additional evidence (scaling, irregular sampling, budget trade-offs, multi-dataset random-sampling comparisons).

Given the above, I recommend weak acceptance. The authors are still encouraged to address the remaining concerns from Reviewer WfpD and QPPP, especially by adding more comprehensive and fair comparison baselines and improving the contextualization with closely related work.

**Reviewer Concerns:**

The novelty of the methodology and some exeperimental details are enhanced by the rebuttal, while some concerns still remain, such as adding more comprehensive and fair comparison baselines and improving the contextualization with closely related work.

**Reviewer Scores:**

Reviewer iw8e (rating 6) and Reviewer KKxL (rating 8) will keep the score.

One of the two reviewers WfpD (rating 4) and QPPP (rating 4) might increase the score.

---

### Decision · Program_Chairs · 2026-01-26

Accept (Poster)